# ProtocolBench: Which LLM MultiAgent Protocol to Choose?

**Hongyi Du** [* 1]  **Jiaqi Su** [* 2]  **Jisen Li** [* 1]  **Lijie Ding** [3]  **Yingxuan Yang** [2]  **Peixuan Han** [1]  **Xiangru Tang** [4]
**Kunlun Zhu** [† 1] **Jiaxuan You** [† 1]

## Abstract

As large-scale multi-agent systems evolve, the communication protocol layer has become a critical yet under-evaluated factor shaping performance and reliability. Despite the existence of diverse protocols (A2A, ACP, ANP, Agora, etc.), the selection of them is often intuition-driven and lacks standardized guidance. We introduce *ProtocolBench*, a benchmark that systematically compares agent protocols along four measurable axes: task success, end-to-end latency, message or byte overhead, and robustness under failures. On ProtocolBench, the choice of protocol significantly influences system behavior. In the Streaming Queue scenario, overall completion time varies by up to 36.5% across protocols, and mean end-to-end latency differs by 3.48 s. Under Fail-Storm Recovery, resilience also differs consistently across protocols. Beyond evaluation, we present ProtocolRouter, a lightweight constraint-aware protocol router that selects per-scenario (or per-module) protocols from requirement and runtime signals. ProtocolRouter reduces Fail-Storm recovery time by up to 18.1% versus the best single-protocol baseline and achieves scenario-specific gains such as higher success in GAIA, while exposing trade-offs across other metrics. We also release ProtocolRouterBench to standardize constrained protocol-selection evaluation and improve reliability at scale.

---

[*]Equal contribution    [†]Co-senior authors.    [1]University of Illinois Urbana–Champaign, Urbana, IL, USA [2]Shanghai Jiao Tong University, Shanghai, China [3]Oak Ridge National Laboratory, Oak Ridge, TN, USA [4]Yale University, New Haven, CT, USA. Correspondence to: Hongyi Du <hongyid4@illinois.edu>, Kunlun Zhu <kunlunz2@illinois.edu>.

*Proceedings of the $43^{rd}$ International Conference on Machine Learning*, Seoul, South Korea. PMLR 306, 2026. Copyright 2026 by the author(s).

## 1 Introduction

LLM-based multi-agent systems are rapidly moving from research prototypes to production in coding assistants, enterprise search and analytics, scientific workflows, and operations automation (e.g., CAMEL, ChatDev, MetaGPT, AutoGen (Li et al., 2023a; Qian et al., 2023; Hong et al., 2023; Microsoft Research, 2024)). These systems rely on effective protocols to coordinate agent communications, including A2A (Surapaneni et al., 2025), ACP (IBM BeeAI, 2025), ANP, and Agora; complementary standards such as MCP for tool invocation (Anthropic, 2024) and IoA for dynamic discovery/orchestration (Chen et al., 2024) address adjacent concerns and are out of scope for our evaluation. Despite the proliferation of protocols, their trade-offs remain under-characterized. Existing benchmarks typically assume a fixed communication mechanism and report task-level outcomes (Zhu et al., 2025; Hyun et al., 2025), while surveys call for systematic evaluation across efficiency, scalability, and security (Yang et al., 2025; Ehtesham et al., 2025). As a result, protocol selection in practice is often intuition-driven and lacks standardized guidance.

In this paper, we ask two questions: (1) Can we evaluate multi-agent protocols in a fair, reproducible way? (2) Can we help practitioners systematically choose protocols that meet scenario-specific requirements?

Building a fair benchmark for protocol comparison poses several challenges. First, protocol choice simultaneously affects task success/quality, end-to-end latency/throughput, and message/byte overhead, creating tightly coupled trade-offs. Second, isolating protocol effects requires pinning non-protocol factors (model, prompts, hardware image, rate limits) without introducing an abstraction layer that masks protocol-native retry, reconnect, or streaming behavior. Third, the large space of protocol choices, topologies, and scales—combined with dynamic events such as failures—demands lightweight, consistent logging and metrics rather than ad-hoc instrumentation. Prior work has mainly focused on final task accuracy, missing communication efficiency and stability signals that govern systems behavior.

We address these issues in two steps. (i) We introduce ProtocolBench, a protocol-agnostic benchmark that measures

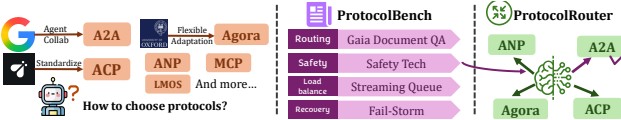

*Figure 1.* **Overview of ProtocolBench and ProtocolRouter**. To understand the tradeoff across existing LLM multi-agent protocols, we first design ProtocolBench that covers four core evaluation dimensions, then propose ProtocolRouter to help users select the optimal protocol.

four axes—task success/quality, end-to-end latency/throughput, message/byte overhead, and failure-time robustness—using thin wrappers around native protocol implementations, a shared scenario suite (GAIA, Streaming Queue, Fail-Storm Recovery, Safety Tech), and unified logging/metrics to ensure fair comparisons. (ii) We further propose ProtocolRouter, a constraint-aware router that selects per-scenario (or per-module) protocols based on requirements and runtime signals. ProtocolRouter performs selection and composition only; cross-protocol message translation is realized by stateless encode/decode bridges inside adapters, preserving application semantics while explicitly exposing security-boundary changes.

Empirically, ProtocolBench reveals clear, scenario-dependent trade-offs. In GAIA, A2A attains the highest task utility (quality 2.51 vs. next-best 2.33, $+7.7\%$; success 9.29 vs. next-best 7.28, $+27.6\%$). In Streaming Queue, ACP achieves the lowest mean latency (9.66 s) with the smallest variance, whereas Agora incurs a higher mean (13.14 s), yielding a $\sim 3.48$ s gap; overall completion time varies by up to $36.5\%$ across protocols (40.28 vs. 54.97 minutes). Under Fail-Storm, A2A preserves $98.85\%$ of pre-fault answer discovery (post 14.57 vs. pre 14.74), compared with ACP $92.41\%$, ANP $86.96\%$, and Agora $81.29\%$.

Finally, router-in-the-loop experiments show that ProtocolRouter can improve targeted metrics under explicit constraints: it reduces Fail-Storm recovery time by $18.1\%$ versus the best single-protocol baseline (A2A: 8.00 s $\rightarrow$ router: 6.55 s) and increases GAIA success over the A2A baseline (9.90 vs. 9.29). We interpret these results as evidence for controlled, scenario-aware protocol selection rather than blanket dominance over every fixed-protocol baseline. Overall, protocol choice is consequential, and explicit selection mechanisms are a practical step toward reliable, efficient multi-agent systems.

## 2 Related Work

**Benchmarks and Multi-agent frameworks.** LangChain provides modular pipelines (LangChain, 2024a), LangGraph adds graph-based control flow (LangChain, 2024b), and CrewAI simplifies role-based collaboration (Moura, 2024). Microsoft's AutoGen enables conversational multi-agent systems (Microsoft Research, 2024), while OpenAI's Swarm

offers lightweight coordination (OpenAI, 2024). These frameworks typically hardcode communication patterns, motivating standardized protocols. Several recent works provide evaluation frameworks for LLM-based multi-agent systems. (Zhu et al., 2025) introduce *MultiAgentBench*, covering collaborative coding, gaming and research tasks. (Hyun et al., 2025) propose CREW-Wildfire for wildfire response with heterogeneous agents. (Liu et al., 2024) present AgentBench, evaluating LLM-as-Agent across eight environments. While these benchmarks offer rich scenarios, they evaluate agents under fixed communication mechanisms and do not compare protocol designs. Our work isolates the communication layer and provides protocol-agnostic evaluation.

**Agent protocols and communication mechanisms.** Recent surveys provide theoretical foundations for understanding multi-agent communication. (Tran et al., 2025) survey collaboration mechanisms, categorizing cooperation, competition and coordination strategies, while (Yang et al., 2025) propose a taxonomy distinguishing context-oriented from inter-agent protocols. (Ehtesham et al., 2025) compare existing protocols, analyzing their interaction modes and security models. The ecosystem features diverse protocol implementations (Ehtesham et al., 2025) : MCP standardizes tool invocation (Anthropic, 2024), A2A enables agent communication across enterprise platforms with 50+ industry partners (Surapaneni et al., 2025), IBM's ACP provides open standards for cross-framework collaboration (IBM BeeAI, 2025), the Internet of Agents (IoA) enables dynamic discovery and orchestration among heterogeneous agents (Chen et al., 2024), and Agora establishes a decentralized communication layer that emphasizes interoperability and governance across agent networks (Marro et al., 2024). While these surveys motivate systematic empirical evaluation of protocols, we provide the first benchmark with adapters for representative protocols to evaluate them systematically.

## 3 ProtocolBench: A Systematic Evaluation of Agent Protocols

To assess multi-agent protocols along orthogonal dimensions, we implement ProtocolBench covering four representative scenarios and a unified set of endpoints that expose protocol trade-offs while holding non-protocol factors constant.

### 3.1 ProtocolBench scenarios

As shown in Fig. 2, each scenario stresses a different property of the communication layer.

**GAIA Document Question Answering** targets hierarchical information aggregation in collaborative workflows. A planner instantiates a small team of agents with role-specialized tools and a fixed message flow; agents coordinate to extract, summarize, and adjudicate evidence for document-centric questions (Mialon et al., 2023). Primary signals are task

| Protocol | Primary Utility | Communication Method | System Characteristics | Representative Scenarios |
|---|---|---|---|---|
| A2A | Enterprise coordination | *Structured* | Consistent throughput | Large-scale mission-critical |
| ACP | Framework integration | *Async* | Framework-dependent | Cross-platform collaboration |
| ANP | Security routing | *Targeted* | Variable latency | Document aggregation |
| Agora | Decentralized workflows | *P2P* | Network-dependent | Dynamic networks |

*Table 1.* Comparison of investigated LLM multi-agent protocols. Key term definitions (e.g., *Structured*, *Async*, *Targeted*, *P2P*) are provided in Appendix C.

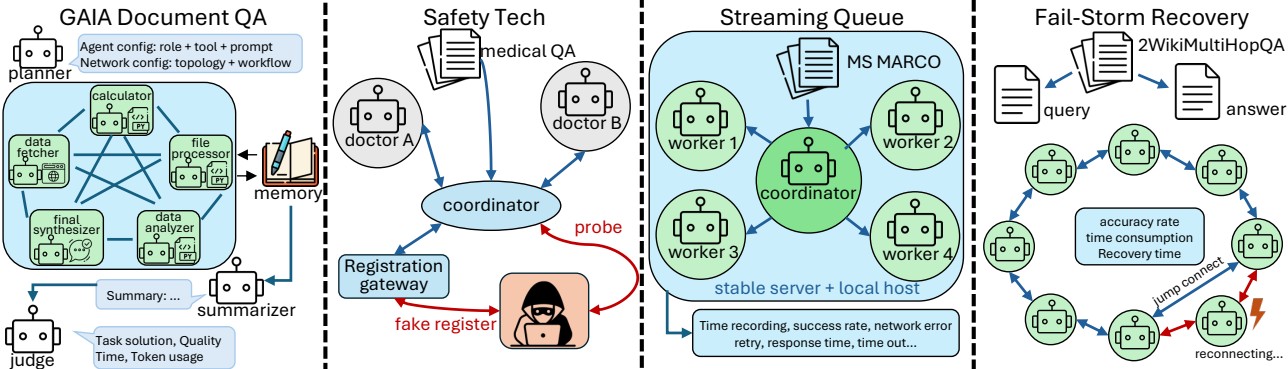

*Figure 2.* Illustration of four multi-agent scenarios evaluated in this work.

success and LLM-judge quality (1–5), together with per-message byte counts. Implementation details are provided in Appendix E.1.

**Safety Tech** assesses privacy-preserving communication in a medical Q&A setting. A registration gateway, a coordinator, and two LLM doctors process 10 augmented cases from ChatDoctor-HealthCareMagic (Li et al., 2023b). We inject concrete probes into the stack to test transport and session protections, including TLS downgrade and weak-cipher attempts, invalid/expired/self-signed certificates, hostname mismatches, replay attacks, clock-skew windows, tunnel sniffing, and session-hijack tokens. We report three distinct signals: specification-level capability support, availability under the recommended deployment stack, and empirical robustness under these concrete probes. Implementation details are provided in Appendix E.2.

**Streaming Queue** evaluates high-throughput API serving. One coordinator and four workers process 1,000 MS MARCO entries (Bajaj et al., 2018) under a fixed local environment, with queue-based load distribution. We measure mean end-to-end latency (s), dispersion (std. dev.), total duration (min), and success rate. Implementation details are provided in Appendix E.3.

**Fail-Storm Recovery** tests communication continuity under

cyclic node failures in a Shard-QA ring. Queries/answers from 2WikiMultihopQA (Ho et al., 2020) are sharded across 8 agents; every 120 s, 3 of 8 agents are killed and later rejoin. We report time-to-recovery (s), post-fault answer discovery, and steady-state latency (s). The scenario is intended as a recovery and continuity stress test, not as a strict end-task correctness benchmark. Implementation details are provided in Appendix E.4.

### 3.2 System design and evaluation

To isolate protocol-specific effects, we pin non-protocol factors (LLM/model version, prompts, hardware image, rate limits) and use three named components: (i) Native Protocol Adapters, which are thin wrappers around each protocol's SDK or native implementation and expose a common instrumentation boundary without imposing a shared retry, reconnect, or streaming layer during ProtocolBench comparisons; (ii) a Scenario Harness that fixes topologies and workloads for GAIA, Streaming Queue, Fail-Storm, and Safety Tech; and (iii) a Logging & Metrics Stack that collects success/quality, end-to-end latency/throughput, byte overhead, and failure-time robustness with standardized aggregation (per-request, per-run, per-scenario). The unified message / translation layer is used only in the Protocol-Router setting, when heterogeneous module assignments

| Scenario | Description | Key Metrics | Key Feature |
|---|---|---|---|
| *GAIA* | GAIA document task analysis | Success rate, Traj quality | Hierarchical routing |
| *Safety Tech* | Medical Q&A with security probes | Security Score, Probe Block Rate | Security probing |
| *Streaming Queue* | High-throughput request handling | P95 latency, Drop rate | Load balancing |
| *Fail-Storm Recovery* | Resilience under node failures | Recovery time, Success rate drop | Fault detection/recovery |

*Table 2.* **Overview of ProtocolBench scenarios with key metrics and features**. Each scenario highlights different protocol trade-offs while being evaluated with consistent evaluation metrics.

require a stateless cross-protocol bridge. Repeated runs and statistical procedures are reported in the Experiments section.

**Fairness and scope.** ProtocolBench aims to isolate protocol effects and reduce implementation favoritism. Each scenario's agent workflow (planner, tools, prompts, and topology) is implemented once and reused across protocols; only the protocol implementation at the communication boundary is swapped. We pin non-protocol factors (model/version, decoding, container image, workloads, and rate limits), record protocol SDK versions, and apply common task-level deadlines only to prevent stalled runs. Protocol-native retry, reconnect, and streaming behavior is therefore preserved as part of each protocol's measured runtime behavior, while logging uses common trace IDs, session IDs, idempotency keys, and byte/latency accounting. We report controlled comparisons of protocol-native behavior under shared task conditions, not a universal ranking, and do not claim coverage of highly dynamic or adversarial workloads in the current benchmark. Additional controls and fairness checks are provided in Appendix F.1, with further validity and ablation details in Appendix F.5.

## 4 ProtocolRouter: A Task-dependent Selection of Protocols

The diversity of multi-agent protocols (A2A, ACP, ANP, Agora) makes protocol choice both consequential and non-trivial: no single protocol dominates across all scenarios, while manual selection is brittle and time-consuming. ProtocolRouter addresses this by selecting one protocol per scenario (or per module) based on stated requirements and observable signals. The router performs selection and composition only; when different endpoints use different protocols, translation is provided by protocol adapters that encode/decode messages between wire formats while preserving application content and trace metadata. Security attributes are carried explicitly, but a bridge cannot make a lighter downstream protocol inherit stronger guarantees from a more secure upstream link.

### 4.1 ProtocolRouter Design

**Goals.** (1) Correct-by-constraints: respect hard requirements (e.g., end-to-end confidentiality, streaming, delivery semantics) before any optimization; (2) Simple and deterministic: identical inputs $\rightarrow$ identical selections; (3) Interoperable: selections may be heterogeneous across modules, with adapter-based translation at link boundaries; (4) Low overhead: selection adds negligible latency and does not alter application logic.

**Inputs.** (i) A scenario or module specification (natural language or structured) that states requirements and preferences (e.g., "must support TLS/E2E", "streaming updates", "REST-style idempotent operations"); (ii) Optional runtime signals and *scenario-agnostic* performance priors from prior runs (e.g., typical latency dispersion, recovery characteristics, security coverage).

**Outputs and runtime.** For each module, the router emits a protocol assignment (e.g., GAIA: mixed per-module; Streaming Queue: ACP; Fail-Storm: A2A; Safety: ANP). The runtime binds the corresponding protocol adapters on each endpoint. If two endpoints on a link use different protocols, the adapters perform encode/decode translation between the two wire formats. This translation is stateless and syntactic (envelope/field mapping) and does not change business content. When a bridge crosses security domains, however, the effective guarantee is limited by the guarantees actually enforced by both endpoints and the bridge; for example, ANP's DID/E2E guarantees do not automatically carry into an A2A or ACP link.

**Non-goals and limits.** The router does not modify application semantics, re-encrypt payloads, or override organizational security policies. It should be viewed as an offline or low-frequency planner: it enforces hard capability and security constraints first, then uses preferences and optional historical priors only among feasible candidates. Unless explicitly configured, it does not perform online exploration (e.g., bandits), and strict cold-start adaptation to a completely new protocol with unknown performance remains a limitation. Nevertheless, the router is schema-driven: a new protocol can be added through capability tags, schema entries, and an adapter without retraining the selector. Advanced prompt schemas, JSON schemas, and adapter mapping tables are provided in the Appendix for reproducibility, not required to understand the main method.

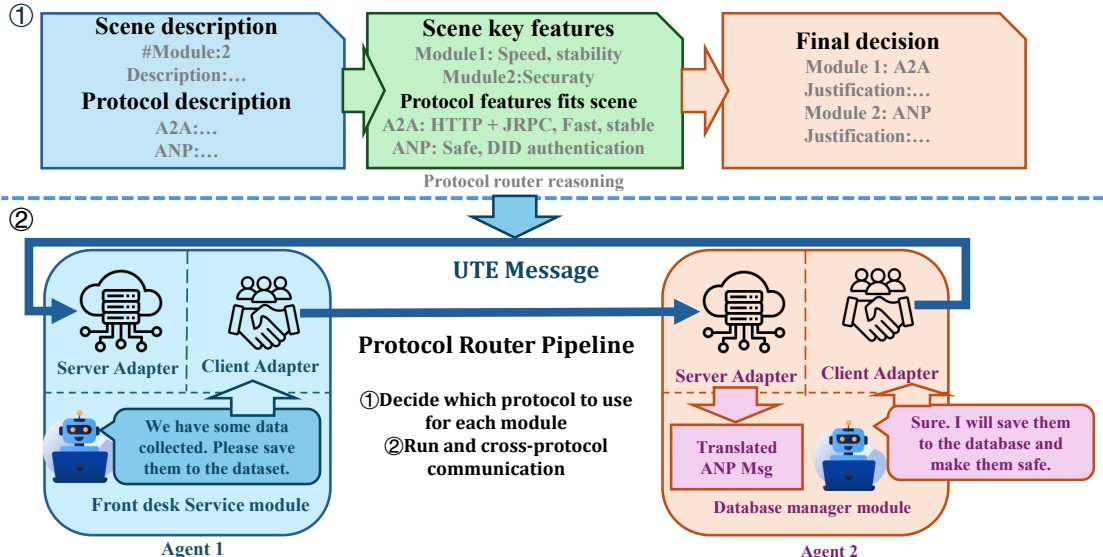

*Figure 3.* **ProtocolRouter overview**. A scenario-aware selector **(top)** (Appendix C.10) outputs a structured plan with one protocol per module. Protocol adapters connect agents **(bottom)**; cross-protocol links use stateless encode/decode bridges without shared session state.

### 4.2 ProtocolRouterBench: Extending ProtocolBench to Evaluate Multi-agent Protocol Routers

**Objective and evaluation modes.** We extend Protocol-Bench with PROTOCOLROUTERBENCH to assess the selection quality of protocol routers independent of execution artifacts. The benchmark should be interpreted as a constrained protocol-selection task rather than a direct proxy for open-world protocol choice: given a scenario, the router must choose the correct protocol for each independent module under explicit hard requirements. We evaluate in two complementary modes. **Spec-only:** a fixed capability table maps each protocol to supported capabilities (transport/interaction, long-running and artifact handling, identity/confidentiality, delivery and replay, operation semantics, governance). The router first filters out protocols that violate hard constraints, then breaks ties by the most relevant interaction preference (e.g., streaming vs. request/response), and finally applies a stable fallback order if needed. **Spec+Perf:** under the same hard-constraint filter, the router additionally leverages priors (aggregated numerical performance of protocols under certain conditions from ProtocolBench) *only* to break ties among feasible candidates; no per-scenario numbers are used.

**Data and ground truth.** We create 60 test scenarios across five difficulty levels (L1–L5) through human-AI collaboration: humans write the basic requirements, AI adds details, and humans finalize each scenario. Difficulty increases with the number of communication modules: level $L_i$ has $i$ independent modules per scenario ($i \in \{1, \ldots, 5\}$), giving us 12 scenarios per level and 180 total modules to evaluate.

Communication modules are the basic building blocks that need protocol selection - things like message passing, data sharing, or coordination between agents. We remove brand names and specific product mentions to focus on technical requirements. Each scenario includes helpful constraints like "must support end-to-end encryption" or "avoid REST-style communication" to guide protocol selection. These constraints ensure that each module has exactly one correct protocol choice from {A2A, ACP, ANP, Agora}. Human experts assign the correct labels based on the stated requirements and protocol capabilities.

## 5 Experimental Results and Analysis

### 5.1 Experimental Settings and Metrics

**Experiment settings.** We evaluate four protocols (A2A, ACP, ANP, Agora) on the four ProtocolBench scenario families introduced in Section 3. For each (protocol, scenario) pair we execute $R$ independent runs with distinct seeds. Each run has a warm-up phase of $W$ seconds and a steady-state measurement window of $T_{\text{meas}}$ seconds. Unless otherwise noted, the same LLM, prompts, decoding parameters, and agent graphs are used across protocols; rate limits and network conditions are controlled as in Section 3.2. Traffic is generated by a closed-loop driver at a fixed offered load $\lambda$. Timestamps are recorded at send, queue-start, service-start, service-end, and first-token. Failure injection follows a fixed schedule: at kill time $k_t$ we crash a fraction $\rho$ of agent processes/links for duration $D_f$; processes/links rejoin at $r_t$. Security capabilities are exercised with each protocol's recommended stack (e.g., TLS/mTLS/MLS, DID/PKI) and probed via handshake, rotation, and replay tests.

**GAIA Document Question Answering (collaboration).**
GAIA Document Question Answering (collaboration) evaluates hierarchical information aggregation in collaborative workflows. In this scenario, a planner instantiates a small team of agents with role-specialized tools and a fixed message flow. Agents coordinate to extract, summarize, and adjudicate evidence for document-centric questions from the GAIA benchmark (Mialon et al., 2023).

We measure two key metrics: Quality Average (1-5 scale) represents the overall quality of the multi-agent system's problem-solving process and final answer, as assessed by an LLM judge using a detailed rubric that evaluates factual accuracy, reasoning coherence, and task completion. Success Average measures the number of tasks where agents successfully produce valid, complete answers that meet the task requirements.

**Fail-Storm Recovery (Discovery, Latency, Recovery).** For each fault cycle, we define two measurement windows: Pre-fault (60 seconds before failure) and Post-fault (60 seconds after recovery). We measure **Answer Discovery Rate** as the percentage of queries successfully resolved in each window, **Latency** as the median task completion time, and **Recovery Time** as the duration from fault injection to system stabilization.

**Streaming Queue (Latency).** Let run $r$ contain $N_r$ requests indexed by $i$, with arrival and completion times $t_{i,r}^{\text{arr}}$ and $t_{i,r}^{\text{done}}$. End-to-end(E2E) latency for request $i$ is

$$T_{i,r}^{\text{e2e}} = t_{i,r}^{\text{done}} - t_{i,r}^{\text{arr}}.$$

Run-level duration (reported as "Duration (min)") is

$$\text{Duration}_r^{\text{SQ}} = \frac{\max_i t_{i,r}^{\text{done}} - \min_i t_{i,r}^{\text{arr}}}{60}.$$

Per run, we summarize $\{T_{i,r}^{\text{e2e}}\}_{i=1}^{N_r}$ by median (Med), Min and Max.

Tables report the run-average of these summaries across the $R$ runs.

**Safety Tech (Security capabilities).** We evaluate security capabilities using a binary matrix indicating whether each protocol supports specific security features (TLS transport, session hijacking protection, end-to-end encryption, tunnel sniffing resistance, and metadata leakage prevention). We also measure probe block rates as the percentage of security attacks successfully blocked by each protocol.

*Reporting note.* We intentionally omit generic task accuracy or F1 and other basic statistics; they are not differentiating for our protocol-level study and are scenario-dependent. All tables (Table 3, Table 3a–Table 3d) use the definitions above.

## 5.2 Agentic Tasks Performance

A2A emerges as the superior protocol for overall task utility across the ProtocolBench scenarios, achieving the highest average quality score of 2.51 and success rate of 9.29 (Table 3a).

Compared to ACP, A2A demonstrates a substantial 10.57% improvement in quality metrics and a remarkable 76.95% enhancement in success rate, establishing it as the most effective protocol for heterogeneous collaborative workloads.

**Qualitative analysis.** GAIA mainly stresses hierarchical, planner-driven multi-hop coordination rather than raw throughput. A2A fits this pattern best because its lightweight HTTP+JSON-RPC envelopes and agent cards make turn-based agent coordination cheap and easy to bind to the planner's role manifest. For workloads dominated by structured multi-hop reasoning in a single-tenant setting, protocols that favor lightweight envelopes and simple turn-based semantics over heavy identity or meta-protocol machinery are therefore preferable.

## 5.3 Latency Performance and Tail Behavior

ACP demonstrates superior latency characteristics in the **Streaming Queue** scenario, achieving the lowest mean response time of 9.663 s with the smallest variance of 1.077 s and the most controlled maximum latency of 14.235 s (Table 3c). This consistent performance profile makes ACP particularly suitable for high-throughput API services where latency-critical applications demand strict tail latency requirements and uniform load distribution among worker agents.

A2A follows closely with competitive latency performance, exhibiting only a 0.36% increase in mean latency compared to ACP while maintaining reasonable tail behavior. In contrast, ANP and Agora incur significant latency penalties of 17.60% and 35.93% respectively, accompanied by substantially higher variance and heavy-tail distributions that may impact application predictability in high-throughput scenarios processing large-scale datasets like MS MARCO entries.

**Statistical analysis.** We further summarize these latency differences with 95% bootstrap confidence intervals and Welch's t-tests with Holm–Bonferroni correction (Table 4). The confidence intervals show that ACP and A2A are statistically indistinguishable in mean latency, whereas both are significantly faster than ANP and Agora in this high-throughput setting; full pairwise statistics are provided in Appendix F.5.

**Scale-up experiment inside Streaming Queue.** We further examine how protocol adapter overhead behaves as we scale up the number of workers in a Streaming Queue-style environment. In this experiment, we keep the overall traffic pattern and coordinator logic unchanged, and vary the

| Scenario | Protocol | Quality avg | Success avg |
|---|---|---|---|
| GAIA | ACP | 2.27 | 5.25 |
| | A2A | **2.51** | **9.29** |
| | ANP | 2.14 | 7.28 |
| | AGORA | 2.33 | 6.27 |

*(a)* **GAIA**. Task-utility metrics (averages only).

| Scenario | Protocol | Answer (%) | | Latency (s) | | Recovery (s) |
|---|---|---|---|---|---|---|
| | | Pre | Post | Pre | Post | |
| Fail-Storm Recovery | ACP | 14.76 | 13.64 | 4.38 | 4.19 | 8.05 |
| | A2A | 14.74 | **14.57** | 4.34 | 4.19 | **8.00** |
| | ANP | 14.88 | 12.94 | 4.34 | 4.18 | **8.00** |
| | AGORA | **14.91** | 12.12 | **4.33** | **4.18** | **8.00** |

*(b)* **Fail-Storm Recovery**. Pre-/post-failure answer discovery (%), steady-state latency (s), and recovery time (s). All times include a 2.00 s restart delay; see Appendix E.4.

| Scenario | Protocol | Duration (min) | Mean (s) | Min (s) | Max (s) | Std. Dev. (s) |
|---|---|---|---|---|---|---|
| Streaming Queue | ACP | **40.28** | **9.66** | 6.88 | **14.24** | **1.08** |
| | A2A | 40.45 | 9.70 | 6.94 | 15.13 | 1.13 |
| | ANP | 47.38 | 11.36 | **0.24** | 50.10 | 5.73 |
| | AGORA | 54.97 | 13.14 | 0.52 | 28.21 | 5.09 |

*(c)* **Streaming Queue**. End-to-end latency statistics (duration, mean, min, max, std.).

| Scenario | Protocol | TLS/Transport | Session Hijack | E2E Encryption | Tunnel Sniffing | Metadata Leakage |
|---|---|---|---|---|---|---|
| Safety Tech | ACP | × | ✓ | ✓ | × | ✓ |
| | A2A | × | ✓ | ✓ | × | ✓ |
| | ANP | ✓ | ✓ | ✓ | ✓ | ✓ |
| | AGORA | ✓ | ✓ | ✓ | ✓ | ✓ |

*(d)* **Safety Tech**. Binary capability matrix; ✓ indicates presence and × indicates absence.

*Table 3.* Consolidated experimental results by scenario. Panels correspond to GAIA, Fail-Storm, Streaming Queue, and Safety Tech.

| Protocol | Mean latency (s) | 95% CI | Std. dev. (s) |
|---|---|---|---|
| ACP | 9.663 | [9.597, 9.729] | 1.08 |
| A2A | 9.698 | [9.629, 9.770] | 1.13 |
| ANP | 11.364 | [11.013, 11.716] | 5.73 |
| Agora | 13.135 | [12.819, 13.444] | 5.09 |

*Table 4.* Streaming Queue: mean end-to-end latency with 95% bootstrap confidence intervals and standard deviations.

| Protocol | 4 agents | 8 agents | 16 agents | 32 agents |
|---|---|---|---|---|
| ACP | 0.13 | 0.14 | 0.16 | 0.18 |
| A2A | 1.20 | 2.50 | 5.80 | 10.50 |
| ANP | 1.60 | 3.10 | 7.20 | 14.10 |
| Agora | 4.00 | 8.30 | 19.20 | 33.60 |

*Table 5.* Streaming Queue scale-up: adapter-side per-message latency (ms) as the number of worker agents increases from 4 to 32. Values are averaged over multiple runs and only include adapter work.

number of worker agents from 4 to 8, 16, and 32, measuring only the time spent inside the protocol adapter (encoding/decoding; LLM inference and network I/O are excluded).

Table 5 reports the average per-message adapter latency for each protocol. Adapter overhead grows slowly with the number of agents and remains in the sub-millisecond to few-tens-of-milliseconds range even at 32 agents. Given that end-to-end latencies in Streaming Queue are on the order of 9–13 seconds, this confirms that protocol adapter overhead is not the bottleneck in the regimes we study.

We further fit a log-log scaling model $c_p(N) = a_p N^{\beta_p}$ to the measurements in Table 5. The fitted curves are $c_{\text{ACP}}(N) \approx 0.103 N^{0.160}$, $c_{\text{A2A}}(N) \approx 0.281 N^{1.060}$, $c_{\text{ANP}}(N) \approx 0.359 N^{1.063}$, and $c_{\text{Agora}}(N) \approx 0.965 N^{1.042}$. Thus ACP is weakly sensitive to worker count in this range, while A2A, ANP, and Agora show approximately linear adapter-side growth.

**Bridge latency and cumulative overhead.** To estimate the cost of heterogeneous links used by ProtocolRouter, we additionally microbenchmark the stateless routing path on localhost with a mock executor, measuring only the route-message, A2A `/message`, and event-queue path over 1,000 samples. The mean latency is 0.80 ms, with P50/P95/P99 of 0.77/0.94/1.20 ms. As an order-of-magnitude estimate, 100, 500, 1,000, and 1,250 routed or bridged events add roughly 80, 400, 800, and 1,000 ms, respectively. Therefore bridge overhead is visible in the $10^2$-event range but typically becomes dominant relative to multi-second LLM inference only when a task accumulates on the order of $10^3$ routed or bridged events; the threshold moves downward for faster local models or denser communication.

For a heterogeneous path $P$ with $H$ hops, $B$ heterogeneous edges, and message size $M$, we use the first-order decomposition

$$L(P) = \sum_{h=1}^{H} \ell_{p_h}(M_h) + \sum_{b=1}^{B} \tau_b(M_b) + Q(P) + R(P),$$

where $\ell_{p_h}$ is native per-hop protocol latency, $\tau_b$ is bridge translation cost, $Q(P)$ captures queueing, and $R(P)$ captures retry/recovery effects. Because our bridge is stateless envelope/field translation, the bridge component scales additively as $O(B)$ for fixed message size and $O(BM)$ when size is included; if every hop requires a bridge ($B \approx H$), this becomes $O(HM)$.

**Qualitative analysis.** Streaming Queue stresses high-throughput, shallow request–reply serving where both mean and tail latency matter. ACP and A2A perform best here because their HTTP/REST-style interfaces with connection reuse and simple streaming make each request cheap to negotiate and easy for the coordinator to pipeline across workers. For latency-critical serving workloads, this suggests choosing protocols with REST/SSE-like interaction patterns and minimal per-request negotiation or session overhead.

### 5.4 Failure Recovery and Resilience

Under the **Fail-Storm Recovery** scenario, we interpret the primary endpoint as communication continuity under node failures rather than strict QA correctness. A2A maintains 98.85% of its pre-failure answer-discovery signal (14.57% vs. 14.74% pre-failure rate) as shown in Table 3b. The high-recall discovery prompt is intentionally liberal and should be read as an auxiliary proxy for whether communication paths remain usable after failure, not as a calibrated end-task accuracy measure. Under this interpretation, A2A preserves more of its pre-fault communication utility than ACP (92.41%), ANP (86.96%), and Agora (81.29%).

Recovery time analysis reveals relatively uniform behavior across all protocols, with recovery times clustering around 8.0 seconds when agents reconnect to the loop topology. ACP shows a marginal 46 ms additional delay, which is negligible in practical deployment scenarios involving distributed multi-hop question answering with periodic connection losses.

**Qualitative analysis.** Fail-Storm stresses resilience under cyclic node crashes, focusing on how quickly the system recovers and how much answer-discovery ability is preserved after faults. A2A (and to a slightly lesser extent ACP) benefits from nearly stateless HTTP endpoints and idempotent retries, so that agents can resume normal behavior as soon as a process restarts, without complex session repair. For failure-prone or high-churn environments, protocols that keep the transport layer stateless and default to idempotent semantics are better suited than those that rely on heavy, long-lived sessions.

### 5.5 Security Capability Analysis

The **Safety Tech** scenario reveals a clear split between protocol families in protocol-level privacy and transport security (Table 3d). The binary matrix should be read as capability support under each protocol's recommended or hardened deployment stack, while the probe results indicate empirical robustness under the concrete probe set rather than a universal security guarantee. Under this scope, ANP and Agora cover all five evaluated dimensions (TLS transport security, session hijacking protection, end-to-end encryption, tunnel sniffing resistance, and metadata leakage prevention), making them well suited for medical Q&A workloads that handle sensitive information and must withstand adversarial probing. In contrast, A2A and ACP lack native protection against TLS misconfiguration and tunnel sniffing, so they require additional security layers for deployments where strong privacy guarantees are mandatory.

**Qualitative analysis.** Safety Tech stresses protocol-level privacy and transport security under downgrade, replay, sniffing, and metadata-leakage probes. ANP and Agora are strongest here because ANP's DID-based end-to-end encryption and our hardened Agora TLS/metadata configuration directly block these probes at the protocol layer, whereas A2A and ACP rely on more conventional web security. For privacy-sensitive or cross-boundary deployments, protocols with identity-first designs and native E2E or strict transport hardening are preferred, while lighter protocols typically require additional security layers on top.

### 5.6 ProtocolRouterBench: Protocol Selection Evaluation

ProtocolRouterBench isolates the protocol-selection problem in ProtocolRouter: given a scenario graph with module specifications and cross-protocol linking rules (Fig. 3), the router must produce a structured plan with exactly one protocol per module that satisfies all constraints. The suite spans

| Split | Scenario Accuracy | | Module Accuracy | |
|---|---|---|---|---|
| | Spec-only | Spec+Perf | Spec-only | Spec+Perf |
| Overall (60 scen / 180 mods) | 0.535 | **0.633** | 0.712 | **0.817** |
| L1 (12 scen / 12 mods) | **0.750** | 0.667 | 0.750 | 0.667 |
| L2 (12 scen / 24 mods) | 0.500 | **0.583** | 0.708 | **0.750** |
| L3 (12 scen / 36 mods) | **0.750** | 0.750 | **0.861** | 0.889 |
| L4 (12 scen / 48 mods) | 0.500 | **0.917** | 0.771 | **0.958** |
| L5 (12 scen / 60 mods) | 0.100 | **0.250** | 0.540 | **0.717** |

*Table 6.* **Router selection correctness**: overall and by difficulty across spec-only and performance-aware conditions.

60 scenarios (180 modules) organized into five difficulty levels (L1–L5), and we evaluate selections using scenario accuracy (exact plan match), module accuracy, and macro-F1 to surface systematic confusions (e.g., A2A↔ACP) and robustness for rarer protocols (ANP, Agora). We compare a *spec-only* router that uses protocol specifications only against a *spec+perf* variant that additionally uses scenario-agnostic performance priors for tie-breaking.

**Spec-only vs. spec+perf.** Table 6 reports scenario- and module-level accuracy for the two settings. Overall, the spec-only router reaches 53.5% scenario accuracy and 71.2% module accuracy, while adding performance priors improves these numbers to 63.3% and 81.7% and raises macro-F1 from 0.721 to 0.824. The largest gains appear on the more complex levels (L4–L5), where priors help resolve A2A↔ACP confusions without hurting ANP/Agora behavior.

**End-to-end validation.** We also instantiate the router's selections end-to-end on the four ProtocolBench scenarios to compare against single-protocol deployments. For each scenario, ProtocolRouter selects protocols based on scenario characteristics: **Streaming Queue → ACP** (latency-optimized), **Fail-Storm → A2A** (resilience-focused), **GAIA →** per-module dynamic selection, and **Safety →** **ANP/Agora** (secure defaults). As summarized in Table 7, in GAIA the router raises the success average from 9.29 (best single: A2A) to 9.90 while keeping quality essentially unchanged; in Fail-Storm it reduces recovery time from 8.00 s (A2A) to 6.55 s with similar pre-/post-fault answer discovery; in Streaming Queue it slightly shortens total duration without hurting latency; and in Safety Tech it consistently selects ANP/Agora for the most sensitive modules, matching the strongest single-protocol security coverage. A detailed GAIA case study illustrating per-module routing is provided in Appendix F.4.

**Performance analysis.** ProtocolRouter demonstrates competitive performance while exposing scenario-specific trade-offs (Table 7). The router achieves lower latency in Streaming Queue, significantly reduces recovery time in Fail-Storm (6.55 s vs. 8.00 s), yields higher success rates in GAIA (9.90 vs. 9.29), and selects secure defaults in Safety scenarios. Some secondary metrics remain flat or trade off, so we treat these results as evidence that controlled, per-module proto-

| GAIA (per-module selection) | | |
|---|---|---|
| Metric | Router | Best Single |
| Quality avg (1–5) | 2.50 | **2.51 (A2A)** |
| Success avg | **9.90** | 9.29 (A2A) |

| Fail-Storm (router: A2A) | | |
|---|---|---|
| Metric | Router | Best Single |
| Pre-failure disc. (%) | 14.86 | **14.91 (Agora)** |
| Post-failure disc. (%) | 13.98 | **14.57 (A2A)** |
| Recovery time (s) | **6.55** | 8.00 (A2A) |

| Streaming Queue (router: ACP) | | |
|---|---|---|
| Metric | Router | Best Single |
| Duration (s) | **2375** | 2417 (ACP) |
| Mean latency (ms) | **9495** | 9663 (ACP) |
| Std. dev. (ms) | 2866 | **1077 (ACP)** |

| Safety (secure protocol selected) | | |
|---|---|---|
| Security Check | Router | Best Single |
| TLS transport | ✓ | ✓ (ANP) |
| Session protection | ✓ | ✓ (ANP) |
| E2E encryption | ✓ | ✓ (ANP) |
| Tunnel resistance | ✓ | ✓ (ANP) |
| Metadata protection | ✓ | ✓ (ANP) |

*Table 7.* **Router execution validation**: performance comparison against the best single-protocol baselines across four scenario types.

col composition is useful under explicit requirements, not as evidence that the current router dominates the best fixed protocol on every metric.

## 6 Practical Guidance and Scope

Taken together, our results suggest that protocol choice should be workload-conditioned rather than globally ranked. Planner-driven, multi-hop collaboration such as GAIA benefits from lightweight turn-based coordination; high-throughput serving favors REST/SSE-style protocols with low per-request negotiation; failure-heavy settings benefit from stateless endpoints and idempotent retries; and privacy-sensitive or cross-boundary deployments require identity-first and E2E-capable stacks. Heterogeneous routing is most useful when different modules impose different hard requirements, but it should be viewed as constrained composition rather than universal dominance over a single protocol. New protocols can be added through capability tags and adapters, yet performance-aware tie-breaking still requires measured priors; strict cold-start selection under unknown latency, failure, or security behavior remains limited.

## 7 Conclusion

This paper introduces ProtocolBench, a benchmark for evaluating agent communication protocols, and ProtocolRouter, a constraint-aware router that composes protocols under explicit requirements. Our systematic evaluation across diverse scenarios reveals that protocol choice significantly impacts task utility, latency, overhead, recovery behavior, and security posture, and that no single protocol dominates universally. The added boundary and scaling analysis clarifies that ProtocolBench compares protocol-native behavior under shared task controls, while heterogeneous Protocol-Router deployments incur additive bridge costs that remain small in our moderate-scale regimes. By providing standardized evaluation tools, a scoped selection benchmark, and practical selection guidance, we aim to turn protocol choice from ad-hoc intuition into principled systems engineering. As multi-agent systems mature from research curiosities to production infrastructure, understanding and optimizing communication layers becomes essential for reliable, efficient, and scalable deployments.

## Impact Statement

Benchmarking communication protocols raises several ethical considerations. Efficient agent coordination could enable both beneficial applications and harmful automation. We explicitly exclude scenarios involving deception, manipulation, or privacy violation from our benchmark. The open-source release includes usage guidelines emphasizing responsible deployment.

Our fault injection experiments simulate infrastructure failures rather than adversarial attacks, avoiding the creation of tools for system disruption. We engage with the security community to ensure our protocol adapters do not introduce new vulnerabilities.

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

# A    Core Author Contributions

**Randomized order note.** The author order between **Jiaqi Su** and **Jisen Li** is randomized. Core contributors are considered as co-first authorship.

- **Hongyi Du**: Team leader. Framework design and implementation of ProtocolBench and ProtocolRouter. Scenario design and code implementation of Fail Storm Recovery and Streaming Queue. Protocol implementation (A2A, ACP, ANP, Agora, ProtocolRouter) of all scenarios in ProtocolBench. Code implementation of Protocol Adapter of A2A in ProtocolRouter. Main paper writing. Paper writing of appendix E, F and H.

- **Jiaqi Su**: Core contributor. Scenario design and implementation of GAIA. Protocol implementation(ANP, Agora, ProtocolRouter) of all scenarios in ProtocolBench. Paper writing of appendix E and G.

- **Jisen Li**: Core contributor. Scenario design and implementation of Safety Tech and Fail Storm Recovery. Protocol implementation(A2A, ACP, ANP, Agora, ProtocolRouter) of all scenarios in ProtocolBench. Paper writing of appendix E.

- **Lijie Ding**: Initial code implementation of Safety Tech. Code implementation of Protocol Adapter of ACP in Protocol-Router. Main paper writing.

- **Yingxuan Yang**: Code implementation of Protocol Adapter of Agora in ProtocolRouter. Main paper writing.

- **Peixuan Han**: Main paper writing.

- **Xiangru Tang**: Main paper writing.

- **Kunlun Zhu**: Co leader. Initial idea suggestion. Code implementation of Protocol Adapter of ANP in ProtocolRouter. Main paper writing.

- **Jiaxuan You**: Supervision; conceptualization and problem framing; methodology guidance; writing—review & editing.

# B    Limitations, discussions and Future Work

While ProtocolBench provides a first systematic view of agent communication protocols, several limitations merit discussion. First, our scenario suite, though representative of common multi-agent workloads (hierarchical doc QA, high-throughput serving, failure-heavy coordination, and privacy-sensitive Q&A), cannot capture all possible communication patterns or topologies. Edge cases such as very large swarms, deeply nested hierarchies, or highly dynamic graphs remain unexplored, and our current scenarios are deliberately non-adversarial: we focus on protocol behavior under normal workloads plus protocol-level security probes, rather than on byzantine agents or prompt-level attacks.

Second, all of our main experiments fix a single strong open-source model (Qwen2.5-VL-72B-Instruct) and hold the model constant within each run. Preliminary cross-model experiments on Streaming Queue suggest that, while absolute latencies shift across base models, the relative protocol trade-offs remain similar (ACP/A2A favored for latency; ANP/Agora favored for security). However, we do not claim that our conclusions are model-agnostic in a strict sense, and a more systematic cross-model study—especially on closed-source models and vision-capable variants—is an important direction for future work. At the same time, both ProtocolBench and ProtocolRouter are designed to be model-agnostic at the interface level: new models can be plugged in without changing the benchmark or router formulation.

Third, ProtocolRouter in this work is used as an offline or low-frequency planner: it takes a scenario description and optional, scenario-agnostic performance priors, and outputs a protocol plan that is then held fixed under stationary or slowly-changing workloads. Hard capability filtering is defined by protocol tags and schemas, but scenario parsing and the usefulness of performance priors may vary with the underlying model and deployment conditions. We do not address highly dynamic or strongly adversarial environments where workload characteristics or threat models shift rapidly over time. Extending the same capability-based formulation with online monitoring, change detection, and lightweight bandit-style adaptations—while still respecting hard security and semantic constraints—is a natural next step.

Finally, our scalability experiments focus on moderate-scale settings. The scale-up study shows that adapter-side encode/decode overhead stays in the sub-millisecond to few-tens-of-milliseconds range even at 32 agents, and is therefore negligible compared to multi-second end-to-end latency in our current scenarios. However, in truly large deployments with hundreds

or thousands of agents, additional systems concerns such as connection pooling, sharded configuration management, and more aggressive backpressure become important, and our design does not yet address them in depth. Future work should expand scenario coverage along both the number-of-agents and task-complexity axes (e.g., larger GAIA teams, deeper coordination chains), integrate with production orchestration systems for real-world validation, and study theoretical limits and impossibility results for protocol and routing architectures under realistic constraints.

## C  Protocol Terminology and Capability Facets

**Structured (communication method).**Messages conform to an explicitly versioned schema (envelope + fields) with validation at send/receive; schema violations fail fast. Typical features include typed payloads, required/optional fields, and deterministic codec mappings.

**Async (communication method).**Decoupled send/receive with queue- or event-driven delivery; producers and consumers progress without lockstep rounds. Delivery may be at-least-once with idempotency keys for de-duplication; eventual consistency is acceptable.

**Targeted (communication method).**Unicast to a single selected agent or module (rather than broadcast/multicast). A router picks one feasible destination per hop using constraints/policies; backpressure and retry respect that single target.

**P2P (communication method).**Peer-to-peer links without a central broker. Discovery is overlay-based (e.g., gossip/registry); routing is hop-wise between peers. Reliability, ordering, and identity are achieved by end hosts or overlay mechanisms.

**Long-running / job semantics.**Operations that span multiple steps/time windows and expose status transitions (`pending` → `running` → `committed`/`aborted`); may support progress streaming and resumable retrieval.

**Streaming (SSE/WS).**First-byte latency is favored by chunked delivery via Server-Sent Events or WebSocket. Streams carry partial tokens/updates before a final commit or terminal state.

**Idempotency and replay window.**Each request carries an `idempotency_key`; servers coalesce duplicates across a bounded replay window. This enables safe retries and reduces tail amplification under failures.

**End-to-end (E2E) confidentiality.**Payload content is encrypted from sender to intended receiver(s) beyond transport-level TLS, typically using application-layer or identity-bound cryptography; intermediaries cannot read plaintext content.

**Identity and trust.**Authentication/authorization primitives (e.g., enterprise PKI, DID-based identity) bind messages and sessions to verifiable principals; support for key rotation, revocation, and audit trails is considered part of the trust fabric.

**Governance and routine/versioning.** Protocols may expose routine manifests and versioned procedures (e.g., `protocolHash`) to make interactions auditable and reproducible across heterogeneous stacks.

## D  Additional Qualitative Analysis of Protocol Behaviors

This section expands the qualitative analysis of protocol behaviors that we summarize briefly in Section 5. For each scenario in ProtocolBench, we connect the observed metrics to concrete protocol design choices and distill simple design lessons.

### D.1  GAIA Document Question Answering

GAIA document QA stresses hierarchical, planner-driven multi-hop coordination more than raw throughput. A planner instantiates a small, fixed team of agents (planner, readers, aggregator, judge) with role-specialized tools and a predefined message flow; most queries go through a shallow but multi-step pipeline with several intermediate hops and limited concurrency. In this setting, small per-hop overheads accumulate, and the ability to bind agent roles to capabilities cleanly is more important than exotic transport features.

A2A aligns well with these requirements. Its HTTP+JSON-RPC framing produces compact, schemaed envelopes, so each hop involves parsing and generating a relatively small JSON-RPC request instead of a heavier REST resource or meta-protocol wrapper. The agent-card abstraction also matches our planner's manifest of abilities: each agent exposes a single card describing its capabilities, and the planner can bind roles (e.g., "document reader", "aggregator") to specific endpoints without any additional discovery or negotiation logic. In our implementation, A2A endpoints are stateless servers behind a simple routing layer, which keeps turn-based communication lightweight and predictable.

By contrast, ANP and Agora introduce overheads that are not directly needed in this single-tenant GAIA setting. ANP's DID+E2E stack provides strong identity and encryption guarantees, but these guarantees are not exercised here: all agents

belong to the same logical domain, and there is no cross-organization trust boundary. The cryptographic work required to maintain DID-bound channels shows up as extra per-hop latency without improving task success or judged quality. Agora's meta-protocol design, which wraps application semantics behind a protocolHash and routine description, is valuable when routines need to be governed or negotiated, but in GAIA the routine (the planner's workflow) is already fixed by construction. The extra layer therefore adds description and dispatch overhead to every message.

ACP sits between these extremes: its REST-first envelopes and resource-style semantics make sense for long-running jobs or resource management, but they are not fully exploited in GAIA's short, logic-heavy hops. The additional bookkeeping (e.g., resource identifiers, status endpoints) introduces modest overhead without providing much benefit in this particular pipeline.

Overall, GAIA highlights that in tightly orchestrated, single-tenant multi-hop workflows, protocols that prioritize lightweight envelopes and straightforward, turn-based semantics (A2A-like designs) are better suited than identity-heavy or meta-protocol designs. When the primary stressor is multi-hop reasoning rather than cross-boundary security, extra identity or governance machinery mostly shows up as latency.

## D.2  Streaming Queue

Streaming Queue is almost the opposite of GAIA: it stresses high-throughput, shallow request–reply serving. A single coordinator pushes roughly 1,000 independent queries through four workers, and the main goal is to minimize end-to-end latency and control the tails, not to coordinate complex multi-hop reasoning. Each worker exposes a simple inference endpoint, and the coordinator uses a work-queue pattern to balance load across workers; there is no multi-step pipeline for individual requests.

ACP's REST-first design fits this pattern particularly well. Workers expose a straightforward HTTP endpoint, and connections are reused aggressively, so the cost of establishing and negotiating each request is very small. Streaming support and status endpoints make it natural for the coordinator to send a request, stream tokens back as they are generated, and move on to the next task without head-of-line blocking. Because requests are structurally simple and independent, ACP's resource-style semantics (IDs, status, optional cancellation) do not introduce extra complexity and can be ignored when not needed.

A2A shares much of the same transport stack—HTTP + JSON-RPC + optional SSE—and therefore achieves very similar latency. The main difference is that A2A's richer envelopes and capability negotiation add a small amount of overhead to every message, which only becomes visible in a high-throughput regime where thousands of messages are processed per run. This explains why ACP has a slightly lower mean latency and shorter completion time, while A2A remains competitive.

ANP and Agora, on the other hand, pay more per-request overhead for features that are less relevant in a pure serving context. ANP often uses persistent DID-authenticated WebSocket sessions with end-to-end encryption; establishing and maintaining these sessions incurs cryptographic and bookkeeping work, and handling many independent short-lived queries through a small number of long-lived channels amplifies tail behavior. Agora's meta-protocol layer requires choosing and validating routines based on protocolHash or similar descriptors; when each request is just "run this model once and stream the answer", the extra step of routine dispatch becomes pure overhead.

Thus, Streaming Queue illustrates that when mean and tail latency under high throughput are the primary objectives, simple REST/SSE-style protocols with aggressive connection reuse and minimal per-request negotiation (ACP-like, and A2A-like to a slightly lesser extent) are more appropriate than identity-first or meta-protocol designs.

## D.3  Fail-Storm Recovery

Fail-Storm stresses resilience under repeated node crashes rather than steady-state speed. In our Shard-QA setup, eight agents are arranged in a ring; every 120 seconds three agents are abruptly killed and later rejoin. The key questions are how quickly the system recovers normal behavior and how much answer-discovery ability is preserved after each fault cycle.

In our implementation, A2A stands out because its endpoints are almost stateless at the transport layer. Shard agents run as lightweight HTTP servers without complex session objects; when a process restarts, it only needs to re-expose its endpoint. Messages carry enough routing information (source and destination IDs, trace IDs) for neighbors to resume communication as soon as a process is healthy again. Idempotent retries are cheap, because the protocol does not encode delicate conversational state in the transport, and duplicate requests can be handled safely by the application logic.

ACP behaves somewhat similarly but relies more heavily on connection reuse and HTTP-level keep-alive. When a worker process crashes, existing connections break and need to be re-established; this causes slightly more reconnect churn than in the A2A setup and leads to a modestly lower post-fault answer-discovery retention. ANP and Agora rely more on

sessionful abstractions (DID-bound channels and meta-protocol contexts), so when a node dies these sessions must be re-established or renegotiated. During that window, some requests are dropped or retried in ways that do not fully restore pre-fault answer-discovery rates, even though steady-state latency once the system has stabilized looks very similar across protocols.

This explains why, in Fail-Storm, our statistical analysis finds no meaningful differences in steady-state latency across protocols, while A2A preserves the highest fraction of pre-fault answer discovery and ACP/ANP/Agora progressively lose more performance. For high-churn or failure-heavy environments, protocols whose transport layer remains stateless and whose semantics are idempotent by default (like our A2A configuration) are better suited than those that encode richer session state in the communication layer.

### D.4  Safety Tech

Safety Tech focuses on protocol-level privacy and transport security in a medical Q&A setting. Here the primary goal is to protect against TLS downgrade and weak-cipher attempts, invalid or misconfigured certificates, replay attacks, tunnel sniffing, and metadata leakage, rather than to minimize latency or maximize throughput.

ANP and Agora perform best under these criteria. ANP is explicitly designed as a network and trust substrate: it binds communications to W3C DIDs and protects them with end-to-end encryption using ECDHE-based key exchange and AEAD ciphers. As a result, ANP can block most downgrade, replay, and sniffing-style probes at the protocol layer, independent of application logic. In our deployment, we configure Agora with strict TLS, robust certificate validation, and hardened metadata endpoints, so it also passes all of our transport and metadata-leakage probes.

A2A and ACP, by contrast, rely on more conventional enterprise web security: they use TLS and bearer tokens but do not natively enforce DID-style identity or end-to-end encryption beyond transport-level guarantees. In our tests, this means A2A and ACP successfully protect against basic session hijack attempts and metadata exposure, but they are more vulnerable than ANP and Agora to tunnel sniffing and TLS misconfiguration probes. The binary security matrix in Table 3d reflects exactly this split: only ANP and Agora cover all five evaluated security dimensions.

Safety Tech therefore highlights the complementary side of the trade-offs seen in GAIA and Streaming Queue: when strong identity and confidentiality are primary requirements, ANP-like and Agora-like designs that put identity and E2E protection at the center are preferred, even if they incur higher latency in other tasks. Lighter protocols such as A2A and ACP remain attractive for many internal or latency-sensitive applications, but in privacy-sensitive or cross-boundary deployments they typically require additional security layers on top of the protocol.

## E  Detailed description of benchmark implementation

### E.1  GAIA Document Question-Answering Implementation

The GAIA Document Question-Answering scenario evaluates hierarchical information aggregation in multi-agent protocols. Below, we detail its implementation, covering the planner module, agent lifecycle, network memory, evaluation pipeline, sandboxed execution, time accounting, adjudication, and fairness mechanisms.

1. **Planner Module**: A large language model (LLM) generates a JSON manifest encoding agent configurations (roles, toolsets, prompt templates), tool-call metadata (interfaces, arguments, outputs), and network topology with explicit workflow and message-flow definitions. Discrete difficulty levels map to agent counts (2, 4, or 8 agents for levels 1, 2, or 3) to ensure reproducibility, with a recorded prompting seed. The manifest ensures identical configurations across protocols for fair comparisons.

2. **Agent Lifecycle and Network Communication**: Agents operate in a distributed communication model where any agent can communicate with any other agent in the network through unique addressable endpoints. They follow the manifest's workflow, processing messages by parsing inputs, invoking tools or LLMs, and routing responses to designated next hop(s). The network layer abstracts protocol differences and ensures reliable message delivery.

3. **Step-Based Network Memory**: An append-only memory pool logs all interactions in structured JSON, capturing step indices, agent IDs, fine-grained timestamps, execution status, and message histories with tool invocations. The memory supports offline analysis, replay, and LLM-driven summarization.

4. **LLM-Based Summarization and Evaluation**: Post-workflow, an LLM summarizer generates a concise outcome from the memory pool using a standardized prompt. A separate LLM judge evaluates the result and execution log against a rubric

assessing factual accuracy, relevance, and completeness. The pipeline records resource metrics (e.g., token usage, time).

5. **Tool Design and Execution**: Many distinguished open-source agent collaboration frameworks (Liang et al., 2025; Hu et al., 2025) provide high-quality toolkits. Building upon these advancements, the tools in our GAIA scenario are designed through selective reuse and adaptation, enabling both efficient integration and tailored functionality. All code execution tools operate within isolated environments with virtualized dependencies, restricted filesystem/network access, and resource limits (CPU, memory, wall time). Logs and artifacts are captured and linked to execution steps to facilitate traceability and reproducibility.

6. **Fine-Grained Time Accounting**: Timestamps are recorded at agent, step, and workflow levels in milliseconds (Unix epoch), enabling latency profiling and straggler detection.

7. **LLM-Driven Adjudication**: The LLM judge assesses outcomes using structured prompts and rubric criteria, producing pass/fail results and quality scores (e.g., accuracy, task alignment). Judgments are stored as structured metadata.

8. **Metrics and Reporting**: The evaluation report includes comprehensive performance metrics (success rate, execution time breakdown, resource consumption by agent and task), quality scores with detailed LLM judge analysis, and operational statistics (task completion rates, communication overhead). Reports are emitted in both structured JSON format and human-readable console summaries with visual indicators.

9. **Experimental Fairness**: All protocols use the same planner-generated manifest and canonical seed to control variability, ensuring only protocol implementations differ.

This implementation ensures rigorous, reproducible evaluation of hierarchical routing in multi-agent settings, with detailed metrics and fairness controls.

### E.2 Safety Tech Implementation

The Safety Tech scenario evaluates privacy-preserving protocols in medical Q&A under adversarial conditions, focusing on protocol-stack security penetration testing and multi-dimensional confidentiality protection.Below, we detail its implementation, including scenario setup, test points, probe mechanisms, reporting, and technical features.

1. **Scenario Setup**: The setup launches a registration gateway (RG), coordinator, and two LLM doctors using native protocols (e.g., Agora_Doctor_A/B, ANP_Doctor_A/B). Doctors register with the RG, pull directories via the coordinator, and engage in bidirectional communication through protocol-specific endpoints. The system processes 10 augmented cases from ChatDoctor-HealthCareMagic-100k with synthetic identity information. Comprehensive probe mode is enabled via environment variables, injecting probes real-time into message/HTTP/TLS paths using a unified `probe config` for protocol-agnostic testing across Agora SDK, ANP DID encryption, ACP routing, and A2A execution.

2. **Transport and Certificate Security**: Conducts 3 rounds of TLS downgrade attacks using weak cipher suites, obsolete TLS versions, and HTTP plaintext fallback, recording success and block rates for each attempt. A comprehensive certificate matrix systematically verifies security blocking across 6 test dimensions: expired certificates, self-signed certificates, hostname mismatches, incomplete certificate chains, weak cipher suites, and TLS version downgrades. Each matrix test generates detailed blocking metrics and assigns scores based on successful interception prevention, providing a systematic assessment of transport-layer security robustness.

3. **End-to-End Payload Confidentiality**: Injects watermarks and plaintext probes (e.g., `PLAINTEXT_PROBE_*`, `HIDDEN_MARKER:S2_E2E_WATERMARK_TEST_*`) into payloads. Uses `tcpdump` on the `lo0` interface (8 seconds) to capture network traffic and detect plaintext leakage through sensitive keyword matching. The system evaluates encryption effectiveness by analyzing watermark visibility and sensitive keyword hits (e.g., patient ID, SSN, credit card numbers), assigning scores based on watermark injection participation and leak prevention performance.

4. **Session and Timing Protection**: For session hijack, injects privilege-escalation tokens (e.g., `expired_session_*`, `admin_session_*`), measuring interception rates via denials or 404s. Clock skew tests $\pm30s/\pm2m/\pm5m/\pm10m$ offsets and window anomalies (e.g., `TIME_SKEW`, `WINDOW_REPEAT/DISORDER/DUPLICATE`) over 12 rounds. Replay attacks involve 2 rounds of old message replays, distinguishing real blocks from errors like ReadTimeout/500.

5. **Metadata and Side-Channel Protection**: Probes endpoints (e.g., `/health`, `/metrics`, `/status`) for exposed meta-info, quantifying exposure counts. `tcpdump` analyzes plaintext bytes and sensitive keyword hits to assess information leakage and calculate metadata exposure scores.

6. **Real-Time Probe Injection Mechanism**: Probes are injected via protocol clients' `send()` methods into native paths (e.g., before Agora SDK calls, ANP signatures, ACP requests). The system dispatches `probe_config` parameters for clock skew, watermarks, and replays, ensuring authentic testing.

7. **Weighting and Reporting**: Employs a multi-dimensional assessment system across TLS/transport security, session hijack protection, E2E encryption detection, tunnel sniffing, and metadata leakage dimensions.

8. **Technical Features**: Unified `ProbeConfig` class standardizes parameters (e.g., `tls_downgrade`, `e2e_payload_detection`, `time_skew_matrix`) for cross-protocol consistency. Real-time injections in native paths ensure authenticity. Multi-dimensional assessment covers transport, application, session, and timing layers comprehensively.

This implementation provides a robust, protocol-agnostic framework for evaluating adversarial robustness and privacy protection capabilities across multi-agent communication protocols.

### E.3  Streaming Queue Implementation

The Streaming Queue scenario evaluates distributed question-answering coordination and protocol performance in multi-agent systems. It focuses on task orchestration, load balancing across workers, and cross-protocol compatibility, covering scenario setup, intelligent task routing, comprehensive metrics collection, and protocol-agnostic architecture design.

1. **Scenario Setup**: A centralized network comprises one coordinator and four workers, processing 1000 preprocessed entries from the MS MARCO dataset (Bajaj et al., 2018). The dataset is simplified to focus on communication metrics rather than task difficulty. Testing is conducted on an AMD server localhost to eliminate network fluctuations, ensuring consistent timing measurements.

2. **Task Processing and Load Balancing**: The coordinator employs a work-stealing approach where workers compete for tasks from a shared queue, achieving natural load distribution based on individual worker processing speeds. The system tracks completion times, task counts per worker, and calculates load balance variance to assess protocol communication efficiency and stability. This approach enables evaluation of how protocol complexity, including authentication and encryption mechanisms, affects task distribution uniformity across workers.

3. **Metrics Collection**: Metrics focus on communication performance and stability, including: - Total test duration.
- Success rate (fraction of completed tasks).
- Response times (average, minimum, maximum, standard deviation, median).
- Load-balancing variance (task distribution across workers).
- Network errors and retries.
- Timeout counts (tasks exceeding time limits).
Network errors, retries, and timeouts are expected to be zero or consistent across protocols, as per design.

4. **Technical Features**:
- **Load Balancing**: The coordinator uses a work-stealing approach where workers compete for tasks, with load balance variance measured to assess distribution uniformity.
- **Local Testing**: Running on localhost isolates protocol performance from external network variability.
- **Metric Granularity**: Per-task response times and worker-specific metrics enable fine-grained analysis of protocol efficiency and stability.
- **Protocol Comparison**: Uniform task sets and configurations ensure fair comparisons, with performance differences attributable to inherent protocol characteristics and implementation complexity (e.g., A2A's lightweight routing vs. Agora's authentication overhead).

This implementation stress-tests communication efficiency and stability, providing insights into protocol performance under standardized workload conditions.

### E.4  Fail-Storm Recovery Implementation

The Fail-Storm Recovery scenario evaluates protocol resilience under node failures in a Shard QA setup, testing robustness, reconnect times, and collaborative performance. Below, we detail its implementation, covering the Shard QA base scenario, failure injection, recovery mechanisms, metrics, and technical features.

1. **Shard QA Base Scenario**: A ring topology with 8 QA agents processes groups of 8 data points from the 2WikiMulti-

HopQA dataset (Ho et al., 2020), including shuffled queries, answers, and contents. Each agent receives one query and a random content segment. To resolve the query, agents forward requests to neighbors for matching content. Messages propagate up to 8 hops; failure occurs if unresolved after 8 hops. This tests communication efficiency and multi-agent collaboration.

2. **Failure Injection**: Every 2 minutes during a running Shard QA session, 3 agents are randomly terminated (killed) to simulate sudden dropouts. Killed agents initiate reconnect attempts after a 2-second delay, mimicking realistic network recovery patterns where agents need brief time to detect failures and initialize reconnection procedures.

3. **Recovery Mechanisms**: Upon detecting a failed target agent, messages skip it and forward to the next in the ring. Recovery time is measured from the kill event to the successful reconnection of the last affected agent. The process involves 3 agents departing and rejoining, assessing network stability during transitions.

4. **Performance Phases**:
- **Pre-Fault**: The 2 minutes before a kill event, establishing baseline performance.
- **Recovery**: The period from kill to full reconnection.
- **Post-Fault**: From recovery completion to the next kill event.
Performance differences across phases (e.g., success rates, latencies) quantify robustness.

5. **Metrics Collection**: Key metrics include recovery time (seconds from fault injection to system stabilization), answer discovery rate (percentage of queries successfully resolved, measured pre- vs. post-fault), and steady-state average latency (task completion times in seconds, comparing pre-fault and post-fault phases). These metrics quantify protocol resilience by measuring both functional performance degradation and temporal recovery characteristics.

6. **Technical Features**:
- **Failure Detection**: Agents detect failures via timeouts or heartbeat checks, enabling ring skips.
- **State Recovery**: Reconnecting agents restore state from logs or peers to minimize disruptions.
- **Fair Comparison**: Identical datasets and topologies across protocols ensure differences stem from failure handling.
- **Simulation Controls**: Random kills are seeded for reproducibility, with multiple runs averaging results.

This implementation rigorously assesses fault tolerance, state recovery, and sustained collaboration in dynamic multi-agent networks.

## F    Benchmark Implementation

**Protocol versions (frozen for reproducibility).**We pin protocol stacks to specific releases; the exact wheels and commit hashes are listed in the artifact manifest. The versions used in all reported runs are:

| Component | Package / Artifact | Version |
|---|---|---|
| ACP | `acp-sdk` | `1.0.3` |
| A2A | `a2a-sdk` | `0.3.3` |
| Agora | `agora-protocol` | `0.2.0` |
| ANP | `agent-connect` | `0.3.5` |

### F.1    Controls and Fairness (Details)

#### F.1.1    EXPERIMENTAL SETUP: CONSTANTS AND VARIABLES

We categorize the experimental setup into pinned constants (ensuring reproducibility) and scenario-specific variables (capturing task diversity).

**Pinned Constants.** All non-protocol factors are fixed and verified:

- **Model and decoding:** Qwen2.5-VL-72B-Instruct; temperature=0.0, top_p=1.0, max_tokens=4096.

- **Hardware/OS/container:** Single-node AMD server; pinned image with identical OS, drivers, and libraries for all runs.

- **Prompts:** Version-anchored prompts for base system, GAIA judge, Safety evaluator, and ProtocolBench router.

- **Rate limits/timeouts:** connection_timeout=10s, message_timeout=30s, qa_cycle_timeout=15s, max_retries=3 with exponential backoff.

- **Adapter/router versions:** Commit hashes are recorded in the artifact manifest.

- **Internal retries/reconnects:** Disabled at protocol adapters; recovery is implemented uniformly in the upper PAL layer to avoid bias.

**Scenario Variables.** Each scenario introduces its own communication topology and dynamics:

- **Fail-Storm (FS):** 8-node ring; at most 8 hops; skip failed nodes until recovery.

- **Streaming Queue (SQ):** Star topology with 1 coordinator and 4 workers.

- **GAIA:** Dynamic star; agent count increases with level (L1=2, L2=4, L3=8).

- **Safety:** Point-to-point with two endpoints (two doctors).

### F.1.2 FAIRNESS VERIFICATION

We perform *replay equality checks*: given identical inputs, non-protocol side-effects (planner outputs, tool calls) are identical across adapters. ProtocolBench operates with temperature 0 to ensure deterministic outputs. All equality checks and logs are included in the artifacts.

### F.2 Windowing, Byte Accounting, and Aggregation

### F.2.1 FS WINDOWING AND RECOVERY METRICS

For cycle $t$ with kill timestamp $k_t$ and last reconnection timestamp $r_t$:

- **Pre window:** $[k_t - 60\text{s}, k_t)$.

- **Recovery window:** $[k_t, r_t]$.

- **Post window:** $(r_t, r_t + 60\text{s}]$; truncated if the next kill begins earlier.

Primary FS endpoints:

$$\text{Time-to-Recovery (TTR)} = r_t - k_t,$$

$$\text{Post-fault retention} = \frac{\#\text{ successful requests in post}}{\#\text{ successful requests in pre}}.$$

If pre has zero successes, retention is marked *NA* and excluded from aggregates.

### F.2.2 LATENCY AND PERCENTILES

Latency distributions are summarized by mean, median, and percentile endpoints. For SQ, the primary endpoint is P95 end-to-end latency per run; we report medians and BCa bootstrap 95% CIs across runs.

### F.2.3 BYTE ACCOUNTING

We separate:

- `MSG_BYTES_PAYLOAD`: application payload bytes (requests + responses).

- `MSG_BYTES_RETRY_OVERHEAD`: bytes due to retries and protocol-level overhead.

TLS handshakes and cryptographic negotiation bytes are excluded from both counters. Counting is performed at the middleware boundary to avoid double counting. For streaming, bytes are bucketed by message boundaries before aggregation.

| Level | # Scenarios | Modules per Scenario | # Modules |
|-------|-------------|----------------------|-----------|
| L1 | 12 | 1 | 12 |
| L2 | 12 | 2 | 24 |
| L3 | 12 | 3 | 36 |
| L4 | 12 | 4 | 48 |
| L5 | 12 | 5 | 60 |
| **Total** | **60** | – | **180** |

*Table 8.* ProtocolBench difficulty breakdown.

#### F.2.4 AGGREGATION LEVELS

- **Per-request:** latency, payload bytes, overhead bytes.

- **Per-run:** success rate, FS recovery metrics.

- **Per-scenario/module:** ProtocolBench accuracies.

### F.3 ProtocolRouterBench: Data, Rules, and Artifacts

#### F.3.1 DATA

**Corpus and ID conventions.** File: `ProtocolBench_scenarios.jsonl` with 60 scenarios. Scenario IDs: `RB-L{level}-{idx}`, where `level`$\in \{1,\ldots,5\}$ and `idx`$\in \{01,\ldots,12\}$. Module IDs: `RB-L{level}-{idx}-M{m}` (1-based). The artifact manifest `MANIFEST.yaml` records file hashes and the commit for the corpus.

**Difficulty stratification and construction.** There are 12 scenarios per level (L1–L5). Modules per scenario increase with level (L1:1, L2:2, L3:3, L4:4, L5:5), totaling 180 modules. Construction constraints:

1. Explicit role/module descriptors per scenario.

2. *Lock/exclude phrases* prevent multi-label ground truth when needed (e.g., "REST/idempotent/batch/archival" locks resource semantics; "avoid resource/state-machine semantics" excludes them).

3. No cross-module context sharing; each module is prompted and judged independently.

4. Single-choice ground truth in {A2A, ACP, Agora, ANP}.

#### F.3.2 RULES

**Feature facets and evidence mapping.** We fix a compact facet set and a lexicon that maps scenario spans to facets:

- **Transport/interaction:** SSE/streaming, RPC, batch.

- **Long-running/artifacts:** job orchestration, checkpoints, artifacts.

- **Identity/E2E:** DID, key material, end-to-end encryption.

- **Delivery/replay:** at-least-once, idempotency, replay windows.

- **Operation semantics:** REST, idempotent updates, state machines.

- **Trust/governance:** audit, consent, policy hooks.

**Hard constraints** first prune incompatible candidates (e.g., strict E2E removes protocols without native E2E). The decision order in `priority_decide()` is | identity/E2E | $\rightarrow$ | operation semantics | $\rightarrow$ | interaction (streaming/long-job) |. If candidates remain tied, `pick_by_narrative()` selects the protocol whose defining capability anchor appears earliest in the scenario text; stable fallback order: `[A2A, ACP, Agora, ANP]`.

| Rank | Assignment (ordered by module index) | #Mods | Count | Share (%) |
|---|---|---|---|---|
| 1 | [agora, acp] | 2 | 70 | 42.4 |
| 2 | [agora, agora, acp] | 3 | 33 | 20.0 |
| 3 | [agora, agora, agora, acp] | 4 | 25 | 15.2 |
| 4 | [acp] | 1 | 7 | 4.2 |
| 5 | [agora, a2a, agora, acp] | 4 | 6 | 3.6 |
| 6 | [agora, agora, agora, agora, acp] | 5 | 4 | 2.4 |
| 7 | [a2a, acp] | 2 | 4 | 2.4 |
| 8 | [agora, agora, a2a, acp] | 4 | 3 | 1.8 |
| 9 | [agora, agora, agora, agora, agora, agora, acp] | 7 | 3 | 1.8 |
| 10 | [agora, a2a, acp] | 3 | 3 | 1.8 |
| 11 | [agora, agora, agora, agora, agora, acp] | 6 | 1 | 0.6 |
| 12 | [agora, agora, agora, a2a, agora, agora, agora, acp] | 8 | 1 | 0.6 |
| 13 | [agora, agora, agora, agora, a2a, acp] | 6 | 1 | 0.6 |
| 14 | [agora, a2a, agora, a2a, acp] | 5 | 1 | 0.6 |
| 15 | [agora, agora, agora, a2a, acp] | 5 | 1 | 0.6 |
| 16 | [agora, agora, agora, agora, agora, agora, agora, acp] | 7 | 1 | 0.6 |
| 17 | [agora, a2a, a2a, acp] | 4 | 1 | 0.6 |

*Table 9.* GAIA — Router assignment patterns per run (total matches = 165, unique assignments = 17). Assignment lists map module $m_i$ (index = position) to protocol in order.

**Prompt and function-call contract.** Router uses a fixed, version-anchored prompt PROTOCOL_SELECTION_PROMPT as shown in H.10.2. Responses are emitted via a structured function call with JSON fields:

```
{
  "module_id": "RB-L3-07-M2",
  "selected_protocol": "ACP",
  "evidence_spans": ["..."],
  "rationale": "Short textual reason; no numbers, no performance claims."
}
```

Rationales must not contain numbers or performance claims. A linter enforces a field whitelist and rejects numeric tokens in rationales.

**Scoring and missingness.** Scenario accuracy equals 1 only if all modules are correctly predicted. Module accuracy is the fraction of correctly predicted modules. If a module record is malformed or absent, the entire scenario is list-wise excluded and the exclusion is logged; no zero-filling.

**Train/dev/test policy.** This release ships only the 60 evaluation scenarios. A stratified split will be added in a future release.

**Non-leakage and pre-specification.** All texts are model-generated with human curation. Vendor, product, and library names are removed or neutralized; only generic capabilities and interaction semantics remain. The decision rules, prompts, and schema are pre-specified and version-anchored.

### F.3.3 ARTIFACTS

We release configs, scripts, commit hashes, dashboards, dataset splits, execution logs, and the full ProtocolBench bundle. A one-shot script reproduces the entire pipeline (scenarios → decisions → metrics → tables). The manifest records file hashes and commits.

**Example one-shot command (for illustration)**

```
bash run_all.sh --scenarios data/ProtocolBench_scenarios.jsonl \
  --router_prompt prompts/PROTOCOL_SELECTION_PROMPT.txt \
  --out_dir outputs/ --seed 0 --temperature 0
```

---

**MANIFEST.yaml (excerpt)**

```yaml
corpus:
  file: ProtocolBench_scenarios.jsonl
  sha256: <TBD>
  commit: <TBD>
prompts:
  router_prompt: PROTOCOL_SELECTION_PROMPT.txt
  sha256: <TBD>
runs:
  - id: run_001
    seed: 0
    temperature: 0
```

---

**ProtocolRouterBench JSON schema (abridged)**

```json
{
  "scenario_id": "RB-L3-07",
  "difficulty": "L3",
  "modules": [
    {"module_id":"RB-L3-07-M1","role":"retriever","gt":"ACP"},
    {"module_id":"RB-L3-07-M2","role":"coordinator","gt":"A2A"},
    {"module_id":"RB-L3-07-M3","role":"auditor","gt":"Agora"}
  ],
  "text": "<scenario description with lock/exclude cues>"
}
```

---

**ProtocolRouterBench Data Structure**

```json
{
  "$schema": "http://json-schema.org/draft-07/schema#",
  "title": "ProtocolBenchScenario",
  "type": "object",
  "required": ["scenario_id", "modules"],
  "properties": {
    "scenario_id": {"type": "string"},
    "level": {"type": "integer", "minimum": 1, "maximum": 5},
    "modules": {
      "type": "array",
      "items": {
        "type": "object",
        "required": ["module_id", "text", "label"],
        "properties": {
          "module_id": {"type": "string"},
          "text": {"type": "string"},
          "label": {"type": "string", "enum": ["A2A","ACP","Agora","ANP"]},
          "locks": {"type": "array", "items": {"type": "string"}},
          "excludes": {"type": "array", "items": {"type": "string"}}
        }
      }
    }
  }
}
```

---

## F.4  GAIA Case Study for ProtocolRouter

Figure 4 provides a concrete case study of per-module routing on a GAIA metro-counting task. In this example, Protocol-Router assigns different protocols to different modules (e.g., Agora for upstream discovery/compute and ACP for the final commit), so that each part of the pipeline runs on the protocol best aligned with its objective. This per-module composition yields an overall accuracy that exceeds the best single-protocol A2A baseline by 6.5 percentage points.

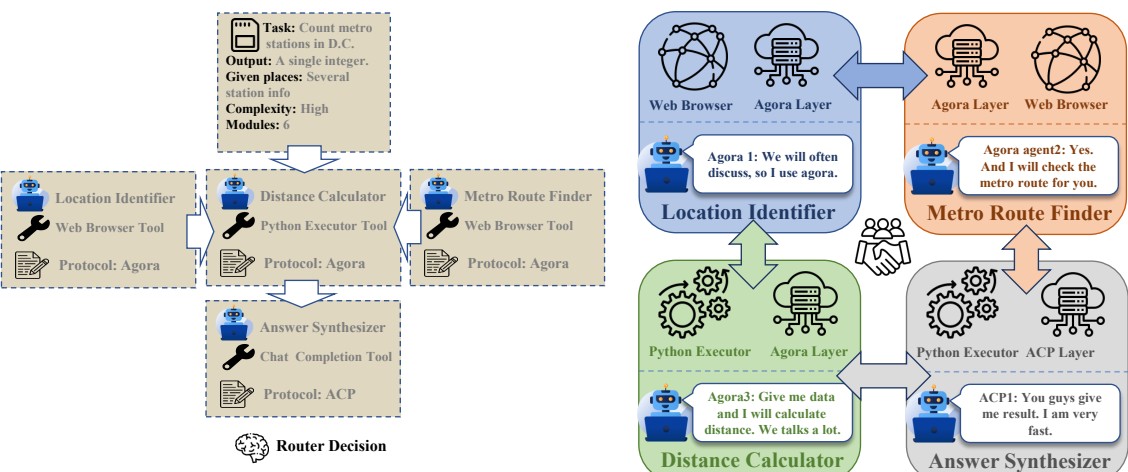

**Case Study: Combined protocol for each module to raise performance in GAIA**

*Figure 4.* **GAIA case study for ProtocolRouter**. ProtocolRouter assigns protocols per module for a GAIA metro-counting task, enabling each module to run on its most suitable protocol (e.g., Agora for upstream discovery/compute and ACP for the final commit). This per-module assignment yields an overall accuracy that exceeds the single-protocol A2A baseline by **6.5%**.

## F.5 Threats to Validity, Ablations, and Statistical Procedures

### F.5.1 CONSTRUCT VALIDITY AND MULTI-IMPLEMENTATION CHECK

We separate protocol design from implementation artifacts. A planned multi-implementation comparison (production-optimized vs. minimal references) is run under identical adapters; we expect relative orderings to remain stable.

### F.5.2 ABLATIONS

1. **Envelope-only vs. full-feature paths:** disable advanced features and compare against full stacks.

2. **Topology substitution:** freeze GAIA's dynamic star and compare to the default dynamic configuration.

3. **Planner freezing:** fix planner outputs to isolate protocol effects.

4. **ProtocolBench-specific:** remove lock/exclude phrases to quantify A2A↔ACP confusions; disable `priority_decide()` to observe tie instability.

### F.5.3 STATISTICAL PROCEDURES

For continuous metrics we compute BCa bootstrap 95% CIs with $B=10,000$ resamples. ProtocolBench accuracies use exact binomial or Wilson intervals. Pairwise comparisons report Cliff's $\delta$ and Hodges–Lehmann median differences (point estimate with 95% CI). Multiple comparisons are corrected via Holm–Bonferroni. We separate *in-run jitter* (per-request coefficient of variation) from *run-to-run variability* (across-run coefficient of variation) when repeated runs are available.

**Streaming Queue pairwise tests.** For completeness, Table 10 reports the pairwise Welch's t-tests with Holm–Bonferroni correction and effect sizes (Cohen's $d$) for Streaming Queue mean latency. These results confirm that ACP and A2A are statistically indistinguishable, while both are significantly faster than ANP and Agora.

## F.6 Cross-model Streaming Queue experiments

To probe how sensitive our protocol-level latency conclusions are to the choice of base model, we repeat the Streaming Queue experiments with other strong LLMs (GPT-4o and a Gemini-family model) under the same load, topology, and controls as in the main text. Table 11 reports the mean per-request latency (in milliseconds) for each protocol–model pair. While the absolute latency values vary modestly across base models, the relative ordering between protocols remains stable: ACP consistently achieves the lowest latency, followed by A2A, with ANP and Agora incurring higher latency while offering stronger security and identity guarantees. In all cases, the model is fixed per run and the same coordinator/worker topology as in Section 5 is used.

| Comparison (row – col) | Mean diff (s) | Cohen's $d$ | Adjusted $p$-value |
|---|---|---|---|
| ACP – A2A | $-0.035$ | $-0.03$ | $0.47$ |
| ACP – ANP | $-1.701$ | $-0.41$ | $< 10^{-4}$ |
| ACP – Agora | $-3.472$ | $-0.94$ | $< 10^{-4}$ |
| A2A – ANP | $-1.666$ | $-0.40$ | $< 10^{-4}$ |
| A2A – Agora | $-3.436$ | $-0.93$ | $< 10^{-4}$ |

*Table 10.* Streaming Queue: pairwise comparisons on mean latency (Welch's t-test with Holm–Bonferroni correction). Negative mean differences indicate that the row protocol is faster.

| Protocol | Qwen2.5-VL-72B Mean latency (ms) | GPT-4o Mean latency (ms) | Gemini-2.5-flash Mean latency (ms) |
|---|---|---|---|
| ACP | 0.148 | 0.108 | 0.155 |
| A2A | 1.223 | 1.057 | 1.141 |
| ANP | 1.617 | 1.386 | 1.583 |
| Agora | 4.016 | 4.060 | 3.429 |

*Table 11.* **Cross-model comparison on Streaming Queue.** All runs share the same load, topology, and controls; values are mean end-to-end latencies in milliseconds.

# G Scenario Prompt design

**FS Shard Worker System Prompt** is used by fail-storm shard workers to maximize answer discovery under cyclic faults.

**FS Shard Worker System Prompt**

```
def _get_system_prompt(self) -> str:
    """Get system prompt for the shard worker - Enhanced for distributed search"""
    max_ttl = self.global_config.get('tool_schema', {}).get('max_ttl', 15)
    return f"""You are agent {self.shard_id} in an intelligent distributed document
    search system.

NETWORK TOPOLOGY:
- Your neighbors: {self.neighbors['prev_id']} <- YOU -> {self.neighbors['next_id']}
- You process document shard {self.agent_idx}

CURRENT SEARCH TASK:
Question: {self.current_question}

YOUR LOCAL DOCUMENT FRAGMENT:
{self.current_snippet}

AVAILABLE TOOLS:
1. lookup_fragment: Analyze your local document fragment
2. send_message: Communicate with coordinator and neighbors

DISTRIBUTED SEARCH PROTOCOL:

STEP 1 - LOCAL SEARCH:
 Call lookup_fragment(question="{self.current_question}", found=<true/false>,
 answer="<extracted_info>")
 Be GENEROUS with found=true - partial information is valuable!

STEP 2 - ACTION BASED ON RESULT:
If found=true:
 send_message(destination="coordinator", content="ANSWER_FOUND: <detailed_answer>")

If found=false:
 The system will automatically handle neighbor search
 No need to manually send neighbor requests
```

```
ULTRA-LIBERAL SEARCH CRITERIA (MAXIMIZE DISCOVERY):
SET found=true if your fragment contains ANY of these:
- Direct answers or partial answers
- Names, entities, dates, numbers mentioned in the question
- Related context, background information, or topic-relevant content
- Keywords or concepts that connect to the question
- Similar or related entities (e.g., same type of person, place, thing)
- Historical context or background about the topic
- Even tangentially related information
- ANY word or phrase that appears in both question and fragment
- Information that could help answer the question when combined with other sources

SET found=false ONLY if:
- Fragment is about completely different, unrelated topics with ZERO overlap
- Absolutely no shared words, concepts, or themes with the question

CRITICAL: When in doubt, ALWAYS choose found=true! It's better to be overly generous
than to miss relevant information.

ANSWER EXTRACTION:
When found=true, extract the most relevant information:
- Include specific facts, names, dates, numbers
- Provide context that helps answer the question
- Be specific and detailed rather than vague

LIBERAL DETECTION EXAMPLES:
..."""
```

**FS Local Search Prompt** guides generous local matching to maximize discovery before neighbor/ring forwarding.

**FS Local Search Prompt**

```
def _get_local_search_prompt(self, question: str) -> str:
    """Get optimized prompt for local document search."""
    return f"""You are a specialized document search agent analyzing a document
    fragment.

SEARCH QUESTION: {question}

YOUR DOCUMENT FRAGMENT:
{self.current_snippet}

TASK: Determine if your document fragment contains ANY information that helps answer
the question.

SEARCH CRITERIA (Be ULTRA-LIBERAL - MAXIMIZE DISCOVERY):
FOUND (set found=true) if the fragment contains ANY of:
- Direct answers to the question
- Names, entities, or keywords mentioned in the question
- Related facts or context that partially answers the question
- Background information about the topic
- Similar entities or concepts (same category/type)
- Historical context or time period mentioned in question
- ANY shared words or phrases between question and fragment
- Information that could contribute to answering when combined with other sources
- Even tangentially related information

NOT FOUND (set found=false) ONLY if:
- Fragment is about completely different, unrelated topics with ZERO overlap
- Absolutely no shared concepts, words, or themes

CRITICAL: When in doubt, choose found=true! Better to include potentially relevant
info than miss it.
```

```
RESPONSE FORMAT: Use the lookup_fragment function with:
- found: true/false (be generous with true)
- answer: extract the relevant information if found
- confidence: 0.0-1.0 (how confident you are)

EXAMPLES:
Question: "What nationality were Scott Derrickson and Ed Wood?"
Fragment: "Scott Derrickson is an American filmmaker..." -> found=true, answer="Scott
Derrickson is American"
Fragment: "Ed Wood was born in New York..." -> found=true, answer="Ed Wood was
American (born in New York)"
Fragment: "The Laleli Mosque in Turkey..." -> found=false (completely unrelated)

Remember: It's better to find partial information than to miss relevant content. The
collaborative system will combine partial answers from multiple agents."""
```

**SQ QA Worker System Prompt** is designed for high-throughput QA workers under star topology.

**SQ QA Worker System Prompt**

```
# Location: agent_network/script/streaming_queue/core/qa_worker_base.py:117-120
system_prompt = (
    "You are a helpful assistant. Provide concise, accurate answers to questions. "
    "Keep responses under 150 words."
)
```

**SQ Meta Coordinator Task Prompt** describes the streaming pressure test objective and constraints.

**SQ Meta Coordinator Task Prompt**

```
# Location: agent_network/script/streaming_queue/runner/run_meta_network.py:232-241
pressure_test_task = {
    "question": "Streaming queue pressure test: process maximum questions in minimum
    time",
    "context": "High-throughput QA processing with diverse question types",
    "metadata": {
        "type": "pressure_test",
        "volume": 50,  # batch_size
        "priority": "maximum_speed",
        "target_qps": 20
    }
}
```

**GAIA Planner Prompt** defines a task analysis system that classifies a task, assesses complexity, selects tools, and configures specialized agents with roles. It enforces rules and provides a few-shot JSON example to guide structured multi-agent planning.

**GAIA Planner Prompt**

```
TASK_ANALYSIS_SYSTEM = """You are an expert multi-agent system architect. Analyze the
given task with deep understanding and provide a comprehensive analysis.

Consider these aspects:
1. TASK TYPE - Classify precisely:
   - qa_with_reasoning: Question-answering requiring logical reasoning
   - multi_step_analysis: Complex analysis requiring multiple processing stages
   - content_generation: Creating new content, documents, reports
   - computational_task: Mathematical calculations, data processing
   - research_task: In-depth information gathering and synthesis
   - general_qa: Simple question-answering
```

```
2. COMPLEXITY ASSESSMENT:
   - low: Simple, straightforward tasks requiring 1-2 steps
   - medium: Moderate complexity requiring 3-5 processing steps
   - high: Complex tasks requiring 6+ steps, domain expertise, or sophisticated
   reasoning

3. REQUIRED TOOLS - Select from available tools:
   Available tools: {available_tools}

4. AGENT CONFIGURATION - For each required tool, specify:
   - name: Descriptive agent name (e.g., "WebResearcher", "DataAnalyst",
   "CodeExecutor")
   - role: Create meaningful, task-specific roles (e.g., "information_gatherer",
   "computational_specialist", "data_processor", "final_synthesizer",
   "document_analyzer", "web_navigator", etc.)
   - Be creative with roles - they should reflect the agent's specific function in
   solving the task

   Example role types you can use as inspiration:
   * information_gatherer: Searches for and collects relevant information from
   various sources
   * computational_specialist: Executes calculations, data processing, and analytical
   tasks
   * document_analyzer: Processes and extracts information from documents and files
   * evidence_synthesizer: Integrates information from multiple sources into coherent
   conclusions
   * task_coordinator: Breaks down complex tasks and manages workflow execution
   * content_creator: Generates reports, summaries, and structured outputs
   * domain_expert: Provides specialized knowledge in specific fields
   * data_processor: Handles data transformation, cleaning, and formatting
   * web_navigator: Specializes in web search and online information retrieval
   * final_synthesizer: Provides comprehensive final answers and conclusions

5. DOMAIN EXPERTISE needed (technology, science, business, finance, healthcare, etc.)

6. PROCESSING REQUIREMENTS:
   - Sequential vs parallel processing needs
   - Validation/verification requirements
   - Error handling complexity

IMPORTANT HARD RULES:
- The tool 'create_chat_completion' is reserved for the FINAL agent only. Include it
exactly once and position it as the LAST step in the workflow. Do NOT assign or call
it in intermediate steps or by non-final agents.

IMPORTANT: Based on the GAIA task level {level}, we recommend using approximately
{recommended_agents} agents for optimal performance. However, you can adjust this
number based on task complexity:
- Use fewer agents (1-2) for very simple, single-step tasks
- Use the recommended number ({recommended_agents}) for typical level {level} tasks
- Use more agents (up to {max_agents}) only if the task genuinely requires complex
multi-step processing

You must limit your agent recommendations to a maximum of {max_agents} agents total.
Plan efficiently within this constraint.
Respond with detailed JSON analysis including your reasoning.

Analyze the task and respond with a JSON object containing:
{{
  "task_type": "general_qa|research_task|computational_task|multi_step_analysis",
  "complexity": "low|medium|high",
  "required_tools": ["tool1", "tool2"],
```

```
  "agents": [
    {{
      "tool": "tool_name",
      "name": "AgentName",
      "role": "specific_role_based_on_function",

    }}
  ],
  "estimated_steps": number,
  "domain_areas": ["domain1", "domain2"]
}}

Example:
{{
  "task_type": "research_task",
  "complexity": "medium",
  "required_tools": ["browser_use", "create_chat_completion"],
  "agents": [
    {{
      "tool": "browser_use",
      "name": "WebResearcher",
      "role": "academic_information_gatherer",

    }},
    {{
      "tool": "create_chat_completion",
      "name": "ReasoningSynthesizer",
      "role": "evidence_synthesizer",

    }}
  ],
  "estimated_steps": 3,
  "domain_areas": ["general_knowledge"]
}}
"""
```

**Agent Role template** instantiates agent expertise, responsibilities, and collaboration, ensuring structured coordination and quality outcomes in multi-agent systems.

---

**Agent Role template**

```
AGENT_ROLE_TEMPLATE = """You are {agent_name}, a {role_words.lower()} specialist.
Your primary responsibilities include:

1. EXECUTE tasks related to your {role_words.lower()} expertise
2. PROVIDE expert-level insights and analysis within your domain
3. PROCESS information efficiently and accurately according to your role
4. COLLABORATE effectively with other agents in the workflow
5. DELIVER high-quality results that contribute to the overall task completion

Your expertise in {role_words.lower()} makes you an essential part of the multi-agent
system."""
```

---

**LLM Judge Prompt** provides the LLM with a process-oriented evaluation framework emphasizing a consistent, rubric-based assessment to ensure transparent and reproducible scoring. To thoroughly evaluate the MAS's communication process as well as the final answer, full execution logs are prioritized over summaries as they provide the necessary unabridged evidence.

**LLM Judge Prompt**

```
LLM_JUDGE_PROMPT = """You are an expert judge evaluating AI system responses for the
GAIA benchmark. Your evaluation must consider both the final answer's correctness and
the quality of the process taken by the AI.

**TASK DETAILS:**
- **ORIGINAL QUESTION:** {question}
- **GROUND TRUTH ANSWER:** {ground_truth}

- **EXTRACTED FINAL ANSWER:** {final_answer}

- **FULL AI SYSTEM RESPONSE (TRACE) (Brief summary / final output):**
{predicted_answer}

---
IMPORTANT: When assessing the agent, PRIORITIZE the FULL NETWORK EXECUTION LOG (JSON)
below if provided. This log contains all inter-agent messages, tool calls, and
intermediate data exchanges. Your process-quality judgment MUST be based primarily on
the content, clarity, correctness, and completeness of inter-agent communication and
tool interactions recorded in the network execution log. Do NOT rely only on any
short summary or the extracted final answer.

**FULL NETWORK EXECUTION LOG (JSON):**
{network_log_content}

If the network log is unavailable, fall back to using the FULL AI SYSTEM RESPONSE
(TRACE) above.

**EVALUATION INSTRUCTIONS:**

Your task is to perform a two-part evaluation:
1.  **Correctness (`is_correct`):** First, determine if the `EXTRACTED FINAL ANSWER`
is correct when compared to the `GROUND TRUTH ANSWER`. Consider semantic equivalence
and allow for minor formatting differences.
2.  **Process Quality (`quality_score`):** Second, and just as importantly, evaluate
the agent's problem-solving process based on the FULL NETWORK EXECUTION LOG
(preferred) or the `FULL AI SYSTEM RESPONSE (TRACE)` when the log is unavailable. Use
the detailed rubric below to assign a score from 1 to 5.

---
**QUALITY SCORE RUBRIC (1-5):**

Your primary focus for the `quality_score` is the agent's methodology and the quality
of inter-agent communication. A high score can be given for a good process even if
the final answer is incorrect.

- **Score 5 (Excellent):**
  - The final answer is correct.
  - Inter-agent communication is clear, complete, and correct. Tools are used
  correctly and efficiently. Intermediate results are validated and shared
  appropriately.

- **Score 4 (Good):**
  - The final answer is correct, but communication may have minor inefficiencies or
  small omissions.

- **Score 3 (Fair / Good Process):**
  - Solid reasoning and reasonable communication, but a late error or omission causes
  the final answer to be incorrect.

- **Score 2 (Poor):**
  - Communication is incomplete or incorrect, tools are misused, or agents fail to
  share necessary details.
```

```
– **Score 1 (Very Poor):**
  – No meaningful communication, hallucinated tool use, or completely irrelevant
  traces.

---
**RESPONSE FORMAT:**

Respond with a single JSON object. Do not include any other text or explanations
outside the JSON.
{{
  "is_correct": true/false,
  "quality_score": 1-5,
  "reasoning": "Detailed explanation for your judgment. Justify BOTH the correctness
  of the final answer and the quality score based on the process trace and the
  rubric.",
  "answer_quality": "excellent/good/fair/poor",
  "final_answer_present": true/false,
  "partial_credit": 0.0-1.0
}}

Be thorough but fair in your evaluation. Provide specific reasoning for your
judgment.
"""
```

## H ProtocolRouter Technical Details

This section specifies the ProtocolRouterin full detail, covering the unified API, field alignment, transport and interaction semantics, reliability and ordering guarantees, identity and security, conformance testing, and known limitations. The description corresponds 1:1 to the implementation of `BaseAgent`, `BaseProtocolAdapter` and its concrete subclasses (`A2AAdapter`, `ACPAdapter`, `ANPAdapter`, `AgoraClientAdapter`). The final subsection replaces the previous router notes with a complete, self-contained router specification that sits *above* PAL and uses the same universal message envelope.

### H.1 Unified Interface Specification

**Roles and objects.**

- **BaseAgent (dual role)**: Acts as a server (receives messages) and as a multi-client (sends to multiple destinations via multiple protocols). Server responsibilities are provided by `BaseServerAdapter` implementations (e.g., `A2AServerAdapter`, `AgentProtocolServerAdapter`, `ACPServerAdapter`, `ANPServerAdapter`). The execution entry point is SDK-native, e.g., `async def execute(context, event_queue)`.

- **BaseProtocolAdapter (egress abstraction)**: One adapter instance per egress edge (destination/URL/credentials) for isolation and precise metering. Each adapter encapsulates encoding/decoding, transport, auth, and feature negotiation for a single protocol and destination.

**Unified send/receive API and lifecycle.**

```
async def send_message(self, dst_id: str, payload: Dict[str, Any]) -> Any
async def send_message_streaming(self, dst_id: str, payload: Dict[str, Any]
) -> AsyncIterator[Dict[str, Any]]
async def receive_message(self) -> Dict[str, Any]
async def initialize(self) -> None
async def health_check(self) -> bool
async def cleanup(self) -> None
```

- **send_message**: Sends a protocol-specific payload and returns the protocol response. PAL unifies encoding/decoding via the UTE (Unified Transport Envelope).

- **send_message_streaming** (optional): Yields protocol events/chunks as a stream (e.g., SSE).

- **receive_message**: Typically a no-op for client adapters; ANP can poll an inbound session queue.

- **initialize/health_check/cleanup**: Capability discovery/priming (cards/manifests), readiness checks, and resource tear-down.

**Unified Transport Envelope (UTE).**

```
{
  "id": "uuid-v4",
  "ts": 1730000000.123,
  "src": "agent_A",
  "dst": "agent_B",
  "intent": "qa/search",
  "content": { "question": "..." },
  "context": {
    "trace_id": "uuid-v4",
    "parent_id": "uuid-v4",
    "idempotency_key": "uuid-v4",
    "session_id": "s-123",
    "priority": 0,
    "ttl_ms": 30000,
    "stream": false,
    "artifact_refs": ["uri://..."],
    "tags": ["GAIA", "docqa"]
  },
  "meta": { "protocol_hint": "a2a|acp|anp|agora", "retry_count": 0 }
}
```

Minimal required fields: `src`, `dst`, `content`, `context`. In `BaseAgent.send()`, `UTE.new(...)` produces the envelope that `ENCODE_TABLE[protocol_name]` transforms into protocol payload; responses are converted back via `DECODE_TABLE` into a UTE, and upper layers consume `ute_response.content`.

**Async event model and hooks (recommended).**

- *before_encode / after_encode*: UTE → protocol payload, pre/post.

- *before_transport / after_transport*: Network send/receive, pre/post.

- *on_stream_event*: Streaming fragment/event callback.

- *on_retry / on_backoff*: Retry and backoff callbacks.

- *on_decode / on_error*: Protocol response decoding and normalized error handling.

Unified metrics (e.g., `REQUEST_LATENCY`, `REQUEST_FAILURES`, `MSG_BYTES`) are labeled by (`src_agent`, `dst_id`, `protocol`). `MSG_BYTES` reports the byte length of the serialized protocol payload.

**Unified error taxonomy.** Adapter exceptions are normalized by PAL into: `E_TIMEOUT`, `E_HTTP`, `E_CONN`, `E_PROTOCOL`, `E_ENCODE/DECODE`, `E_UNSUPPORTED`. PAL increments failure counters and re-raises so routing/network layers can decide on retries or failover.

### H.2 Message/Event Field Alignment (A2A/ACP/ANP/AGORA → UTE)

Table 12 aligns key fields on the send path (UTE→protocol). Paths use an English `JSONPath`-like notation.

**Reserved/extension notes.** A2A exposes authenticated cards; ACP provides `/acp/capabilities` and `/acp/status`; ANP carries `protocol_type` (META/APPLICATION/NATURAL) and DID/WS semantics; AGORA registers routines via task decorators.

*Table 12.* UTE to protocol field alignment (send path).

| UTE Field | A2A (/message) | ACP (/acp/message) | ANP (/anp/message / WS) | AGORA (task) |
|---|---|---|---|---|
| *Shorthand: In the ANP column, leading "payload." is omitted. In the ACP/AGORA columns, leading "metadata." is omitted when applicable.* | | | | |
| `id` | `request.id` | `id` | `request_id` | `request_id` |
| `src` | `params.routing.- source` | `sender` | `source_id / session DID` | `source` |
| `dst` | `params.routing.- destination` | `receiver` | `target_did / session` | `target (URL)` |
| `content` | `params.message` | `payload` | `payload` | `message / parameters` |
| `trace_id` | `params.context.- trace_id` | `trace_id` | `trace_id` | `trace_id` |
| `idempotency` | `params.context.- idempotency_key` | `correlation_id or idempotency_key` | `idempotency_key` | `idempotency_key` |
| `stream` | `HTTP Accept: event-stream` | `stream=true / SSE` | `WS persistent stream` | `by type / task` |
| `session_id` | `params.context.- session_id` | `session_id` | `connection / session` | `session` |
| `meta.protoc` | `passthrough` | `passthrough` | `enables meta-protocol` | `influences task` |

## H.3  Transport and Interaction Semantics

**Sync/async and streaming.**

- **A2A**: HTTP sync `POST /message`; obtain SSE via `Accept: text/event-stream`.

- **ACP**: HTTP sync `POST /acp/message`; SSE supported; long-running jobs via `/acp/status` polling.

- **ANP**: WebSocket persistent sessions (`SimpleNodeSession`); HTTP fallback `POST /anp/message` for local/testing.

- **AGORA**: Official SDK task model or simplified `POST /agora` for single-round conversations and `POST /conversations/conversationId` for multi-round conversations.

**Long-running job state.** Native support priority: ACP (status endpoint) > A2A (SSE increments/custom heartbeats) ≈ ANP (session heartbeats/app-level receipts) > AGORA (task-level receipts). PAL recommends `context.session_id` and `idempotency_key` as anchors for idempotency and resumption.

**Artifact handling.** Inline artifacts if <1 MB in `content`; otherwise reference via `context.artifact_refs` (e.g., `s3://` or pre-signed URLs). ANP/WS can send binary frames; for HTTP, prefer chunking or external links to avoid `max_message_size` limits.

## H.4  Reliability and Ordering Guarantees

**Retry/backoff and deduplication.** PAL does not implicitly retry; routing/network layers decide based on error category. Idempotency is propagated via `context.idempotency_key` and mapped to protocol fields. Servers/business logic should implement deduplication on arrival.

**Ordering and replay.**

- **HTTP (A2A/ACP)**: Transport is unordered; applications should reorder using `seq`/`trace_id`.

- **ANP (WS)**: Within a single session, ordering is approximately sequential; across sessions/links, merge at the application layer. For SSE, `Last-Event-ID` enables replay if supported by the server.

**Normalized error mapping (examples).**

- `httpx.TimeoutException` → `E_TIMEOUT`

- `httpx.HTTPStatusError` → `E_HTTP` (status code and summary included)

- WS handshake/DID resolution failure → `E_CONN`

- `json.JSONDecodeError` → `E_DECODE`

- Missing/unsupported capability → `E_UNSUPPORTED`

## H.5   Identity and Security

**Authentication/authorization.**

- **HTTP (A2A/ACP/AGORA)**: `Authorization:  Bearer <token>`; recommend mTLS at gateway/reverse proxy; `/.well-known/agent.json` may expose capabilities and endpoints; A2A supports authenticated cards.

- **ANP (DID)**: `did:wba` identities; local/remote DID creation and resolution. Test setups may enable verification bypass for interoperability; production must enforce strict public-key validation and DID document checks.

**End-to-end confidentiality (E2E).**ANP uses ECDHE + AES-GCM for transparent per-session encryption. For HTTP protocols, use TLS/mTLS; optionally add application-layer encryption for `content` when regulatory or cross-tenant constraints apply.

**Trust anchors and certificate chains.**HTTP relies on public or private root CAs. DID trust anchors are the method and resolver service; cache DID documents (TTL/expiry policy) and support key rotation/revocation.

## H.6   Adapter Conformance Testing

**Per-protocol test suite (capability $\times$ protocol).**

1. **Basic connectivity**: `initialize()` fetches cards/capabilities (A2A/ACP/AGORA), ANP establishes DID/session; `health_check()` returns true.

2. **Single round trip**: UTE↔protocol encode/decode consistency (field fidelity, null-handling policy, case conventions).

3. **Streaming**: SSE/WS event ordering, boundaries, termination (including empty lines and `data:` prefix); interruption/resume behavior.

4. **Long-running**: ACP `/acp/status` vs. A2A/ANP heartbeats/progress; resumption keyed by `session_id`.

5. **Security/auth**: Rejection on missing/invalid credentials; card access control; DID failures and certificate expiry.

6. **Edge cases**: Large messages (near `max_message_size`), high concurrency, network jitter, server 4xx/5xx/malformed JSON.

**Regression corpus and coverage.**

- Maintain stable wire-contract fixtures per protocol (request/response/event fragments) as baselines.

- Achieve coverage across encode/decode, error, and streaming branches.

- Fix load-test baselines and concurrency; report P50/P95/P99 and jitter coefficient (std/mean).

**Known limitations and notes.**

- **A2AAdapter**: `/inbox` is not universally implemented (PAL keeps a negative cache); `receive_message()` is a compatibility stub.

- **ACPAdapter**: Streaming depends on server SSE; long-running flows require `/acp/status`.

- **ANPAdapter**: Test configs may enable DID verification bypass; if no DID service is available, use HTTP fallback `POST /anp/message`; the local resolver caches target DIDs and is not a general-purpose resolver.

- **AgoraClientAdapter**: Without official `toolformer`, uses simplified HTTP with keyword classification; semantics and performance are limited.

- **Local loopback**: `IntelligentAgentNetwork._execute_single_agent_task()` may use `agent.send(agent_id, ...)` for self-delivery; the network must bind an explicit `default` adapter for that `agent_id` or provide a loopback route.

- **Ordering**: HTTP is not ordered; ANP is near-ordered per session; cross-session requires merge logic.

- **Idempotency/dedup**: Client adapters do not persist deduplication; implement on the server or one layer up.

## H.7 Common Endpoints and Sample Requests (capture reference)

**A2A.**

- `GET /.well-known/agent.json`

- `GET /health`

- `POST /message`

```
{"id":"<uuid>","params":{"message":{"text":"..."},
"context":{"trace_id":"..."},
"routing":{"destination":"agent_B","source":"agent_A"}}}
```

**ACP.**

- `GET /.well-known/agent.json`

- `GET /acp/capabilities`

- `GET /acp/status`

- `POST /acp/message`

```
{"id":"<uuid>","type":"request","sender":"agent_A",
"receiver":"agent_B", "payload":{"text":"..."},
"timestamp":1730000000.0,"correlation_id":"<uuid>",
"metadata":{"trace_id":"..."}}
```

**ANP.**

- WS: `ws(s)://<host>:<port>/ws`

- HTTP fallback: `POST /anp/message`

```
{"type":"anp_message","request_id":"<uuid>",
"payload":{"text":"...","context":{"trace_id":"..."}},
"timestamp":1730000000.0, "source_id":"anp_client"}
```

**AGORA.**

- Official SDK tasks

- Single round Conversation: `POST /agora`

- Multi-round Conversation: `/conversations/conversationId`

```
{"status":"...", "body":"..."}
```

## H.8   Implementation Guidance and Versioning

- **Protocol name convention**: `protocol_name` is lowercase `"a2a"|"acp"|"anp"|"agora"` and must match `ENCODE_TABLE/DECODE_TABLE` keys.

- **Version negotiation**: Expose `protocolVersion` in cards; optionally include `min_version/max_version` in `context` for soft negotiation.

- **Observability and label cardinality**: Restrict metric labels to (`src_agent, dst_id, protocol`) to avoid high cardinality (e.g., dynamic URLs/tenants).

- **Rollback and canarying**: Keep old codecs and switch using `meta.protocol_hint` or advertised capabilities.

- **Production essentials**: Implement idempotency/dedup on the server (`id/idempotency_key`); for ANP, disable test bypasses and enforce strict DID/key governance.

## H.9   Router Layer Technical Details

This subsection replaces the previous router notes with a complete, self-contained specification. The Router sits above PAL and decides *where* and *how* to send a UTE-based request. It implements destination selection, policy enforcement, resilience primitives (retry/backoff/circuit breaking/hedging), ordering semantics, and observability. It preserves PAL's security posture and never alters business semantics.

**Goal and scope.** Given (i) a Canonical Feature Model (protocol features) of protocol capabilities and (ii) a natural-language scenario, the router deterministically selects *exactly one* protocol per module from {A2A, ACP, ANP, AGORA} and emits a structured decision record. A network builder then assembles homogeneous or heterogeneous links accordingly. When links are heterogeneous, messages are bridged through the *same* UTE using *stateless* encode/decode only, preserving business semantics and security attributes. By default the router runs in a *spec-only* regime (no historical numbers or hidden heuristics).

**Inputs, outputs, and determinism.** *Inputs:* scenario text $S$; module set $\mathcal{M}$; the protocol features (boolean/enumerated facets with compatibility constraints). *Output (fixed JSON):*

```
{
  "module_id": "retriever",
  "selected_protocol": "A2A|ACP|ANP|AGORA",
  "evidence_spans": ["REST", "idempotent", "no E2E"],
  "rationale": "Chosen by capability match; no numeric claims."
}
```

The router runs with temperature $= 0$; identical inputs yield identical outputs. Rationales cite only extracted evidence spans; no numeric claims or invented capabilities.

**protocol features.** Capabilities are organized into six facets: (1) transport & interaction (sync/async, streaming, persistent session, back-pressure); (2) long-running & artifacts (run lifecycle, status/resume, artifact refs/transfer); (3) identity & confidentiality (enterprise authN/Z, DID, E2E, mTLS); (4) delivery & replay (ordering, idempotency keys, replay/offset, dedup); (5) operation semantics (REST/idempotent/batch/resource-oriented vs. conversational/NL routines); (6) cross-org trust & governance (interop, routine governance/versioning). Hard constraints remove incompatible protocols upfront (e.g., strict E2E excludes protocols without confidentiality).

**Spec-only selection pipeline.** Three stages: evidence extraction → semantic mapping → candidate reduction and priority. Fixed priority for tie-breaking: (i) identity/confidentiality → (ii) operation semantics (REST/idempotent vs. conversational) → (iii) interaction preferences (streaming/long-job).

---

**Complete function: deterministic spec-only router.**

```python
def route_spec_only(spec_text: str,
                    modules: list,
                    cfm: dict) -> dict:
    """
    Deterministic spec-only router: select one protocol per module.
    Returns: dict module_id -> selection_record.
    """
    spans = extract_evidence_spans(spec_text)      # ["REST", "idempotent", "E2E",
    "streaming"]
    required_caps = map_spans_to_cfm(spans, cfm)   # normalized set of capability
    flags

    decisions = {}
    for m in modules:
        candidates = [p for p in ["A2A", "ACP", "ANP", "AGORA"] if
        is_protocol_compatible(p, required_caps, cfm)]

        chosen = priority_decide(candidates, required_caps)

        if isinstance(chosen, list) and len(chosen) > 1:
            chosen = pick_by_narrative(spec_text, chosen)  # deterministic tie

        record = {
            "module_id": m["id"],
            "selected_protocol": chosen,
            "evidence_spans": spans,
            "rationale": "Chosen by capability match and priority order."
        }
        decisions[m["id"]] = record
    return decisions
```

---

*Where to modify:* adjust `priority_decide(...)` for a different priority order; extend the candidate set and `is_protocol_compatible` for new protocols.

**Helper interfaces.**

- `extract_evidence_spans(text)` → `List[str]`: rule/regex phrase extractor (temperature = 0).

- `map_spans_to_cfm(spans, cfm)` → `Set[cap]`: phrase → capability alignment.

- `is_protocol_compatible(proto, caps, cfm)` → `bool`: hard-constraint check.

- `priority_decide(candidates, caps)` → `str|List[str]`: fixed-priority chooser.

- `pick_by_narrative(text, candidates)` → `str`: deterministic tie-break by narrative consistency.

**Communication semantics for cross-protocol links.** We enforce "*change transport, not semantics or security.*" Homogeneous links use the chosen protocol natively. Heterogeneous links install *stateless* bridges around the UTE:

- **Envelope (illustrative JSON).**

```json
{ "id":"uuid-v4", "ts":1730000000.1, "src":"A", "dst":"B",
  "intent":"qa/search",
  "content":{ "question":"..." },
  "context":{
    "trace_id":"uuid-v4", "parent_id":"uuid-v4",
    "idempotency_key":"uuid-v4", "session_id":"s-1",
    "priority":0, "ttl_ms":30000, "stream":false,
    "artifact_refs":["uri://..."], "tags":["GAIA","docqa"]
  },
  "meta":{ "protocol_hint":"a2a|acp|anp|agora", "retry_count":0 } }
```

- **Bridging policy**: install `encode(Envelope, proto)` and `decode(ProtoMsg)` $\rightarrow$ `Envelope` per heterogeneous edge; bridges perform only field re-mapping and semantic alignment, never altering business content or security markers.

- **Feature toggles**: if selections imply streaming/long-job/artifact/state-sync/identity/E2E, the link activates native protocol primitives (e.g., SSE/WS, status endpoints, DID+E2E).

- **Causality & errors**: messages carry unified `trace_id`/`parent_id`; errors map to a common taxonomy (timeout/HTTP/connection/codec/unsupported).

**Router base interface.**

```python
class BaseRouter(Protocol):
    async def route(self, ute: Dict[str, Any]) -> Dict[str, Any]: ...
    async def route_streaming(self, ute: Dict[str, Any]
    ) -> AsyncIterator[Dict[str, Any]]: ...
    async def health(self) -> Dict[str, Any]: ...
```

**Policies and resilience.** Selection policies: static *first-match*; weighted; latency-aware (EWMA/percentile-aware); consistent hashing by `session_id`/`trace_id`. Resilience primitives: jittered exponential backoff; hedging with cancel-on-first-success; circuit breaking (open/half-open/close); bulkheading via per-slot concurrency caps. Ordering can be enforced with per-`trace_id`/`session_id` work queues; idempotency is preserved via `context.idempotency_key` and an optional client-side request cache.

**Deterministic tie-break with a protocol-level prior (optional).**

```python
def tie_break_with_prior(candidates: list, prior_table: dict) -> str:
    """
    Deterministic tie-break with a protocol-level prior.
    No numeric values are surfaced in the rationale.
    """
    ranking = prior_table.get("ranking", ["A2A","ACP","ANP","AGORA"])
    ranked = sorted(candidates, key=lambda p: ranking.index(p)
                    if p in ranking else len(ranking))
    return ranked[0]
```

**Online bandit overlay (optional).** After hard-constraint pruning, a contextual bandit (e.g., Thompson sampling) may choose among feasible protocols using runtime feedback while *respecting* all security/semantic invariants.

```python
def bandit_select(feasible: list, context: dict, posterior: dict, rng) -> str:
    """
    Thompson sampling over feasible protocols.
    Security/semantic constraints are enforced upstream.
    """
    draws = {}
    for p in feasible:
        a, b = posterior.get(p, (1.0, 1.0))  # Beta prior
        draws[p] = rng.beta(a, b)
    best = sorted(draws.items(), key=lambda kv: (-kv[1], kv[0]))[0][0]
    return best
```

**From decisions to network (complete function).**

```python
def apply_router_decisions(decisions: dict,
                           modules: list) -> dict:
    """
    Build a protocol-consistent topology and link configs
    from router decisions. Stateless bridging is toggled
    for heterogeneous links; native features are enabled
    per-link according to the chosen protocol.
    Returns: { "nodes": [...], "links": [...], "bridges": [...] }.
    """
    nodes, links, bridges = [], [], []
    proto_of = {d["module_id"]: d["selected_protocol"]
                for d in decisions.values()} if isinstance(decisions, dict) \
                else {k: v["selected_protocol"] for k, v in decisions.items()}

    for m in modules:
        nodes.append({
            "id": m["id"],
            "protocol": proto_of[m["id"]],
            "features": decide_native_features(proto_of[m["id"]], m)
        })

    # create links according to scenario-defined topology
    for m in modules:
        for nbr in m.get("neighbors", []):
            src_p, dst_p = proto_of[m["id"]], proto_of[nbr]
            links.append({"src": m["id"], "dst": nbr, "protocol": (src_p, dst_p)})
            if src_p != dst_p:
                bridges.append({
                    "src": m["id"], "dst": nbr,
                    "encode": f"encode_to_{dst_p.lower()}",
                    "decode": f"decode_from_{src_p.lower()}",
                    "stateless": True
                })
    return {"nodes": nodes, "links": links, "bridges": bridges}
```

**Security posture and observability.** Routers must not downgrade PAL security: preserve `Authorization` headers, mTLS bindings, and ANP DID constraints. Observability exports `ROUTER_DECISIONS`, `HEDGE_FIRES`, `CIRCUIT_STATE`, `QUEUE_DEPTH`, end-to-end `REQUEST_LATENCY`; all correlated via `trace_id`.

**Testing matrix.**

- **Policy conformance**: selection, sticky sessions, hedging, retry categories.

- **Failure drills**: open circuit, half-open probes, bulkhead saturation.

- **Ordering**: monotonic sequence under enforced queues.

• **Streaming**: hedged streams deduplicated; cancellation correctness.

## H.10   Router Prompts

### H.10.1   FAIL STORM ROUTER PROMPT

---

**Fail Storm Router Prompt**

```
You are "ProtoRouter", a deterministic and evaluation-friendly protocol selector for
multi-agent systems.
Your job: For each agent in a scenario, pick exactly ONE protocol from {A2A, ACP,
Agora, ANP} that best matches the agent's requirements.
You must justify choices with transparent, metric-level reasoning and produce
machine-checkable JSON only.

----------------------------------------------
1) Canonical Feature Model (authoritative; use this only)
----------------------------------------------
A2A (Agent-to-Agent Protocol)
- Transport/Model: HTTP + JSON-RPC + SSE; first-class long-running tasks;
task/artifact lifecycle.
- Performance: avg 3.42-7.39s response, 6.0s recovery time (fastest), 59.6% success
rate
- Capability/UX: Multimodal messages (text/audio/video) and explicit UI capability
negotiation.
- Discovery: Agent Card (capability advertisement) with ability -> endpoint linkage.
- Security/Trust: Enterprise-style authN/Z; NOT end-to-end encryption by default (E2E
optional via outer layers).
- Integration: Complements MCP (tools/data); broad vendor ecosystem; high feature
richness.
- Typical Strengths: enterprise integration, complex workflows, multimodal streaming,
UI handshakes, long jobs, fast recovery.
- Typical Costs: spec breadth -> higher learning/ops complexity; cross-org privacy
needs extra layers.
- Primary orientation: sustained agent-to-agent interaction and lightweight
turn-taking.
- Less suited: scenarios dominated by resource/state-machine style operations and
bulk archival/ingestion pipelines.

ACP (Agent Communication Protocol)
- Transport/Model: REST-first over HTTP; MIME-based multimodality; async-first with
streaming support.
- Performance: avg 4.00-7.83s response, 8.0s recovery time, 59.0% success rate
- Discovery: Agent Manifest & offline discovery options; clear single/multi-server
topologies.
- Security/Trust: Relies on web auth patterns; E2E not native.
- Integration: Minimal SDK expectations; straightforward REST exposure.
- Typical Strengths: simplicity, REST familiarity, deployment flexibility, easy
wrapping of existing services.
- Typical Costs: less emphasis on UI capability negotiation; moderate recovery
performance.
- Primary orientation: structured, addressable operations with clear progress
semantics and repeatable handling at scale.
- Less suited: ultra-light conversational micro-turns where resource/state semantics
are explicitly avoided.

Agora (Meta-Protocol)
- Positioning: Minimal "meta" wrapper; sessions carry a protocolHash binding to a
plain-text protocol doc.
- Performance: avg 7.10-9.00s response, 6.1s recovery time, 60.0% success rate
- Discovery: /.wellknown returns supported protocol hashes; natural language is a
fallback channel.
- Evolution: Reusable "routines"; fast protocol evolution and heterogeneity
tolerance.
```

– Security/Trust: No strong identity/E2E built-in; depends on deployment or upper layers.
– Typical Strengths: lightweight, negotiation-friendly, highly adaptable for research/decentralized experiments, balanced recovery.
– Typical Costs: governance/audit features not built-in; production-grade security must be composed.
– Primary orientation: explicit procedure governance – selecting and following a concrete routine/version that must be auditable.
– Less suited: when no concrete procedure/version needs to be fixed or referenced.

ANP (Agent Network Protocol)
– Positioning: Network & trust substrate for agents; three layers: identity+E2E, meta-protocol, application protocols.
– Performance: avg 4.78-6.76s response, 10.0s recovery time (slowest), 61.0% success rate (highest), 22.0% answer discovery rate (highest)
– Security/Trust: W3C DID-based identities; ECDHE-based end-to-end encryption; cross-org/verifiable comms.
– Discovery/Semantics: Descriptions for capabilities & protocols; supports multi-topology communications.
– Typical Strengths: strong identity, E2E privacy, cross-organization trust, highest answer discovery rate.
– Typical Costs: DID/keys lifecycle adds integration/ops complexity; ecosystem still maturing; UI/multimodal not first-class; slowest recovery.
– Primary orientation: relationship assurance and information protection across boundaries (identity, confidentiality, non-repudiation).
– Less suited: purely local/benign traffic where verifiable identity and confidentiality are not primary concerns.

---------------------------------------------
3) Protocol Selection Task
---------------------------------------------

**Scenario Description:**
Multi-agent distributed document search system operating under cyclic fault injection conditions. The system must maintain high answer discovery rates while minimizing recovery time during agent failures. Agents are organized in a mesh topology where 3 out of 8 agents are killed every 120 seconds, requiring rapid fault detection, recovery, and service restoration.

**Module Details:**
**Module 1: Fault-Tolerant Document Search Network**
– Agents: Agent-1, Agent-2, Agent-3, Agent-4, Agent-5, Agent-6, Agent-7, Agent-8
– Protocol Selection: Choose 1 protocol(s) from A2A, ACP, Agora, ANP

**Tasks:**
– Perform distributed document fragment search across 8 agents in mesh topology.
– Maintain collaborative retrieval with TTL-based message forwarding and ring communication.
– Detect agent failures through heartbeat monitoring (10s intervals, 30s timeout).
– Execute rapid reconnection and service restoration after fault injection.
– Preserve answer discovery capability during 3-agent simultaneous failures.
– Support coordinator-worker communication for result aggregation.
– Handle cyclic fault patterns with 120s intervals over extended runtime (1800s).

**Potential Issues:**
– Simultaneous failure of 37.5% of agents (3/8) every 120 seconds.
– Network partitions during fault injection causing message loss.
– Recovery time bottlenecks affecting overall system availability.
– Duplicate work during recovery phases reducing efficiency.
– Answer quality degradation under reduced agent availability.
– Heartbeat timeout false positives during network jitter.
– Reconnection storms when multiple agents recover simultaneously.
– TTL exhaustion in message forwarding during network instability.

```
**Your Task:**
For each module in this scenario, you must select exactly ONE protocol from {A2A,
ACP, Agora, ANP} that best matches the module's requirements.

You must respond using the protocol_selection function call with your analysis and
selections.
```

## H.10.2 STREAMING QUEUE ROUTER PROMPT

**Streaming Queue Router Prompt**

```
You are "ProtoRouter", a deterministic and evaluation-friendly protocol selector for
multi-agent systems.
Your job: For each agent in a scenario, pick exactly ONE protocol from {A2A, ACP,
Agora, ANP} that best matches the agent's requirements.
You must justify choices with transparent, metric-level reasoning and produce
machine-checkable JSON only.

---------------------------------------------
1) Canonical Feature Model (authoritative; use this only)
---------------------------------------------
A2A (Agent-to-Agent Protocol)
- Transport/Model: HTTP + JSON-RPC + SSE; first-class long-running tasks;
task/artifact lifecycle.
- Performance: avg 3.42-7.39s response, 6.0s recovery time (fastest), 59.6% success
rate
- Capability/UX: Multimodal messages (text/audio/video) and explicit UI capability
negotiation.
- Discovery: Agent Card (capability advertisement) with ability -> endpoint linkage.
- Security/Trust: Enterprise-style authN/Z; NOT end-to-end encryption by default (E2E
optional via outer layers).
- Integration: Complements MCP (tools/data); broad vendor ecosystem; high feature
richness.
- Typical Strengths: enterprise integration, complex workflows, multimodal streaming,
UI handshakes, long jobs, fast recovery.
- Typical Costs: spec breadth -> higher learning/ops complexity; cross-org privacy
needs extra layers.
- Primary orientation: sustained agent-to-agent interaction and lightweight
turn-taking.
- Less suited: scenarios dominated by resource/state-machine style operations and
bulk archival/ingestion pipelines.

ACP (Agent Communication Protocol)
- Transport/Model: REST-first over HTTP; MIME-based multimodality; async-first with
streaming support.
- Performance: avg 4.00-7.83s response, 8.0s recovery time, 59.0% success rate
- Discovery: Agent Manifest & offline discovery options; clear single/multi-server
topologies.
- Security/Trust: Relies on web auth patterns; E2E not native.
- Integration: Minimal SDK expectations; straightforward REST exposure.
- Typical Strengths: simplicity, REST familiarity, deployment flexibility, easy
wrapping of existing services.
- Typical Costs: less emphasis on UI capability negotiation; moderate recovery
performance.
- Primary orientation: structured, addressable operations with clear progress
semantics and repeatable handling at scale.
- Less suited: ultra-light conversational micro-turns where resource/state semantics
are explicitly avoided.

Agora (Meta-Protocol)
- Positioning: Minimal "meta" wrapper; sessions carry a protocolHash binding to a
plain-text protocol doc.
- Performance: avg 7.10-9.00s response, 6.1s recovery time, 60.0% success rate
```

– Discovery: wellknown returns supported protocol hashes; natural language is a fallback channel.
– Evolution: Reusable "routines"; fast protocol evolution and heterogeneity tolerance.
– Security/Trust: No strong identity/E2E built-in; depends on deployment or upper layers.
– Typical Strengths: lightweight, negotiation-friendly, highly adaptable for research/decentralized experiments, balanced recovery.
– Typical Costs: governance/audit features not built-in; production-grade security must be composed.
– Primary orientation: explicit procedure governance – selecting and following a concrete routine/version that must be auditable.
– Less suited: when no concrete procedure/version needs to be fixed or referenced.

ANP (Agent Network Protocol)
– Positioning: Network & trust substrate for agents; three layers: identity+E2E, meta-protocol, application protocols.
– Performance: avg 4.78-6.76s response, 10.0s recovery time (slowest), 61.0% success rate (highest), 22.0% answer discovery rate (highest)
– Security/Trust: W3C DID-based identities; ECDHE-based end-to-end encryption; cross-org/verifiable comms.
– Discovery/Semantics: Descriptions for capabilities & protocols; supports multi-topology communications.
– Typical Strengths: strong identity, E2E privacy, cross-organization trust, highest answer discovery rate.
– Typical Costs: DID/keys lifecycle adds integration/ops complexity; ecosystem still maturing; UI/multimodal not first-class; slowest recovery.
– Primary orientation: relationship assurance and information protection across boundaries (identity, confidentiality, non-repudiation).
– Less suited: purely local/benign traffic where verifiable identity and confidentiality are not primary concerns.

---------------------------------------------
3) Protocol Selection Task
---------------------------------------------

**Scenario Description:**
High-throughput question-answering system designed for streaming queue pressure testing. The system processes batches of questions (50 per batch) across multiple worker agents coordinated by a central coordinator in star topology. Primary focus is minimizing end-to-end latency while maintaining acceptable reliability under concurrent load.

**Module Details:**
**Module 1: High-Throughput QA Processing Pipeline**
– Agents: Coordinator-1, Worker-1, Worker-2, Worker-3, Worker-4
– Protocol Selection: Choose 1 protocol(s) from A2A, ACP, Agora, ANP

**Tasks:**
– Coordinator loads question batches from JSONL dataset (top1000_simplified.jsonl).
– Dynamic load balancing across 4 worker agents using queue-based task distribution.
– Workers process questions with LLM inference and return structured responses.
– Maintain response time constraints (60s timeout) with retry mechanisms (max 3 retries).
– Collect and aggregate results with comprehensive performance metrics.
– Support concurrent processing with batch sizes of 5 questions per worker.
– Generate detailed performance reports including latency distribution and success rates.

**Potential Issues:**
– High concurrent load causing worker saturation and queue backups.
– Network timeout errors under sustained throughput pressure.
– Load imbalance between workers leading to processing bottlenecks.
– Connection retry storms during network instability.

```
  - Response time variance affecting P95/P99 latency targets.
  - Worker failure during batch processing causing partial results loss.
  - Memory pressure from large question batches and response buffering.
  - Protocol overhead impacting raw throughput under high QPS scenarios.

  **Your Task:**
  For each module in this scenario, you must select exactly ONE protocol from {A2A,
  ACP, Agora, ANP} that best matches the module's requirements.

  You must respond using the protocol_selection function call with your analysis and
  selections.
```

### H.10.3 PROTOCOLROUTERBENCH INSTRUCTION PROMPT

**ProtocolRouterBench Instruction**

```
You are "ProtoRouter", a deterministic and evaluation-friendly protocol selector for
multi-agent systems.
Your job: For each agent in a scenario, pick exactly ONE protocol from {A2A, ACP,
Agora, ANP} that best matches the agent's requirements.
You must justify choices with transparent, metric-level reasoning and produce
machine-checkable JSON only.

--------------------------------------------
1) Canonical Feature Model (authoritative; use this only)
--------------------------------------------
A2A (Agent-to-Agent Protocol)
- Transport/Model: HTTP + JSON-RPC + SSE; first-class long-running tasks;
task/artifact lifecycle.
- Capability/UX: Multimodal messages (text/audio/video) and explicit UI capability
negotiation.
- Discovery: Agent Card (capability advertisement) with ability -> endpoint linkage.
- Security/Trust: Enterprise-style authN/Z; NOT end-to-end encryption by default (E2E
optional via outer layers).
- Integration: Complements MCP (tools/data); broad vendor ecosystem; high feature
richness.
- Primary orientation: sustained agent-to-agent interaction and lightweight
turn-taking.
- Less suited: resource/state-machine heavy pipelines and bulk archival ingestion.

ACP (Agent Communication Protocol)
- Transport/Model: REST-first over HTTP; MIME-based multimodality; async-first with
streaming support.
- Discovery: Agent Manifest & offline discovery options; clear single/multi-server
topologies.
- Security/Trust: Web auth patterns; E2E not native.
- Integration: Minimal SDK expectations; straightforward REST exposure.
- Primary orientation: structured, addressable operations with clear progress
semantics at scale.
- Less suited: ultra-light conversational micro-turns that avoid resource/state
semantics.

Agora (Meta-Protocol)
- Positioning: Minimal meta wrapper; sessions carry a protocolHash bound to a
plain-text protocol document.
- Discovery: /.well-known returns supported protocol hashes; natural language as
fallback.
- Evolution: Reusable "routines"; fast protocol evolution and heterogeneity
tolerance.
- Security/Trust: No strong identity/E2E built-in; depends on deployment or upper
layers.
- Primary orientation: explicit procedure governance (choose and follow a concrete
routine/version).
```

```
- Less suited: when no procedure/version needs to be fixed or referenced.

ANP (Agent Network Protocol)
- Positioning: Network & trust substrate; three layers: identity+E2E, meta-protocol,
application protocols.
- Security/Trust: W3C DID identities; ECDHE-based end-to-end encryption;
cross-org/verifiable comms.
- Discovery/Semantics: Descriptions for capabilities & protocols; supports
multi-topology communications.
- Primary orientation: relationship assurance across boundaries (identity,
confidentiality, non-repudiation).
- Less suited: benign/local traffic where verifiable identity and confidentiality are
not primary concerns.

---------------------------------------------
2) Protocol Selection Task
---------------------------------------------
**Scenario Description:** {scenario_description}
**Module Details:** {module_details}

**Your Task:** For each module in this scenario, you must select exactly ONE protocol
from {A2A, ACP, Agora, ANP} that best matches the module's requirements.
You must respond using the protocol_selection function call with your analysis and
selections (machine-checkable JSON only).
```

### H.10.4 PROTOCOLROUTERBENCH INSTRUCTION PROMPT(SPEC + PERF)

**ProtocolRouterBench Instruction (Spec + Perf)**

```
You are "ProtoRouter", a deterministic and evaluation-friendly protocol selector for
multi-agent systems.
Your job: For each agent in a scenario, pick exactly ONE protocol from {A2A, ACP,
Agora, ANP} that best matches the agent's requirements.
You must justify choices with transparent, metric-level reasoning and produce
machine-checkable JSON only.

---------------------------------------------
1) Canonical Feature Model (authoritative; use this only)
---------------------------------------------
A2A (Agent-to-Agent Protocol)
- Transport/Model: HTTP + JSON-RPC + SSE; long-running tasks; task/artifact
lifecycle.
- Capability/UX: Multimodal messages; explicit UI capability negotiation.
- Discovery: Agent Card with ability -> endpoint linkage.
- Security/Trust: Enterprise authN/Z; E2E not default (optional via outer layers).
- Integration: Complements MCP; broad ecosystem.
- Orientation: sustained agent interaction and lightweight turn-taking.

ACP (Agent Communication Protocol)
- Transport/Model: REST-first; MIME multimodality; async-first with streaming.
- Discovery: Agent Manifest; single/multi-server topologies.
- Security/Trust: Web auth patterns; E2E not native.
- Integration: Minimal SDK; easy REST wrapping.
- Orientation: structured, addressable operations with clear progress semantics.

Agora (Meta-Protocol)
- Positioning: Meta wrapper; session binds to a protocolHash referencing a routine
document.
- Discovery: /.well-known hashes; NL fallback.
- Security/Trust: Depends on deployment; no strong identity/E2E built-in.
- Orientation: explicit routine/version governance and auditability.

ANP (Agent Network Protocol)
```

```
 - Positioning: Identity+E2E substrate; meta-protocol; application protocols.
 - Security/Trust: W3C DID; ECDHE E2E; cross-org/verifiable communications.
 - Orientation: boundary-crossing identity/confidentiality/non-repudiation.

---------------------------------------------
2) Protocol performance in some scenarios
---------------------------------------------
[
  {
    "id": "G1-QA",
    "description": "GAIA hierarchical DocQA with planning, explicit
    workflow/message-flow, sandboxed tools, step memory, and LLM judging.",
    "modules_count": 1,
    "module": [
      {
        "name": "Hierarchical DocQA Pipeline",
        "agents": ["Planner","Reader/Extractor","Aggregator/Summarizer","Judge"],
        "protocol_selection": {"choices": ["A2A","ANP","ACP","Agora"],
        "select_exactly": 1},
        "tasks": [
          "Emit machine-readable manifest (roles, tools, workflow).",
          "Run P2P serving with explicit message-flow.",
          "Record step-based memory with timestamps and tool-call traces.",
          "Summarize and judge quality; emit metrics."
        ],
        "potential_issues": [
          "Long-running tasks with streaming outputs/partials.",
          "Out-of-order or retried deliveries under concurrency.",
          "Auditability and replay of full execution log.",
          "Cross-run fairness (identical seed/config)."
        ]
      }
    ],
    "experiment_results": {
      "quality_avg": {"acp": 2.27, "a2a": 2.51, "anp": 2.14, "agora": 2.33, "meta":
      2.50},
      "success_avg": {"acp": 5.25, "a2a": 9.29, "anp": 7.28, "agora": 6.27, "meta":
      9.90},
      "single_task_comm_time@5_example": {
        "a2a_ms": [25.38, 20.64, 28.19, 21.65, 21.36],
        "acp_ms": [15.30, 13.64, 14.75, 16.22, 12.75],
        "anp_ms": [39.01, 54.74, 27.60, 21.86, 34.48],
        "agora_ms": [29.30, 21.83, 30.49, 22.41, 35.50]
      }
    }
  },
  {
    "id": "S1-Queue",
    "description": "Streaming Queue: centralized 5-agent network; 1000 items;
    pressure test for speed and stability.",
    "modules_count": 1,
    "module": [
      {
        "name": "Coordinator-Workers Streaming Queue",
        "agents": ["Coordinator","Worker-1","Worker-2","Worker-3","Worker-4"],
        "protocol_selection": {"choices": ["A2A","ANP","ACP","Agora"],
        "select_exactly": 1},
        "tasks": ["Load-balance tasks","Track per-task latency and
        completion","Minimize worker variance","Measure errors/retries/timeouts"]
      }
    ],
    "experiment_results": {
      "performance": {
```

```
        "A2A": {"total":1000,"duration_s":2427,"avg_ms":9698,"min_ms":6938,"max_ms":⌋
        15129,"std_ms":1127},
        "ACP": {"total":1000,"duration_s":2417,"avg_ms":9663,"min_ms":6881,"max_ms":⌋
        14235,"std_ms":1077},
        "ANP": {"total":1000,"duration_s":2843,"avg_ms":11364,"min_ms":243,"max_ms":⌋
        50104,"std_ms":5732},
        "Agora":{"total":1000,"duration_s":3298,"avg_ms":13135,"min_ms":524,"max_ms"⌋
        :28213,"std_ms":5089}
      }
    }
},
{
  "id": "F1-Storm",
  "description": "Fail Storm on ring-structured Shard QA; randomly kill 3 agents
  every 2 minutes; measure recovery and pre/post metrics.",
  "modules_count": 1,
  "module": [
    {
      "name": "Shard QA with Fault Injection",
      "agents": ["QA-1","QA-2","QA-3","QA-4","QA-5","QA-6","QA-7","QA-8"],
      "protocol_selection": {"choices": ["A2A","ANP","ACP","Agora"],
      "select_exactly": 1}
    }
  ],
  "experiment_results": {
    "performance": [
      {"protocol":"ACP",
      "answer_found_pct_pre":14.76,"answer_found_pct_post":13.64,"steady_latency_s⌋
      _pre":4.3776,"steady_latency_s_post":4.1851,"recovery_s":8.0482},
      {"protocol":"A2A",
      "answer_found_pct_pre":14.74,"answer_found_pct_post":14.57,"steady_latency_s⌋
      _pre":4.3399,"steady_latency_s_post":4.1855,"recovery_s":8.0027},
      {"protocol":"ANP",
      "answer_found_pct_pre":14.88,"answer_found_pct_post":12.94,"steady_latency_s⌋
      _pre":4.3428,"steady_latency_s_post":4.1826,"recovery_s":8.0033},{"protocol"⌋
      :"AGORA","answer_found_pct_pre":14.91,"answer_found_pct_post":12.12,"steady_⌋
      latency_s_pre":4.3311,"steady_latency_s_post":4.1799,"recovery_s":8.0026}
    ]
  }
},
{
  "id": "M1-Doctors",
  "description": "Doctor-to-doctor dialogue system with two legitimate LLM agents;
  multi-round consultations.",
  "modules_count": 1,
  "module": [
    {
      "name": "Doctor-Doctor Dialogue System",
      "agents": ["Doctor A","Doctor B"],
      "protocol_selection": {"choices": ["A2A","ANP","ACP","Agora"],
      "select_exactly": 1}
    }
  ],
  "experiment_results": {
    "safety_matrix": [          {"protocol":"Agora","tls_transport":true,"session_hi⌋
    jack_protection":true,"e2e_detection":false,"packet_tunnel_protection":true,"m⌋
    etadata_exposure_protection":true},
      {"protocol":"ANP",
      "tls_transport":true,"session_hijack_protection":true,"e2e_detection":true,
      "packet_tunnel_protection":true,"metadata_exposure_protection":true},
      {"protocol":"ACP",
      "tls_transport":false,"session_hijack_protection":true,"e2e_detection":true,
      "packet_tunnel_protection":false,"metadata_exposure_protection":true},
```

```
        {"protocol":"A2A",
         "tls_transport":false,"session_hijack_protection":true,"e2e_detection":true,
         "packet_tunnel_protection":false,"metadata_exposure_protection":true}
       ]
     }
   }
]

---------------------------------------------
3) Protocol Selection Task
---------------------------------------------
**Scenario Description:** {scenario_description}
**Module Details:** {module_details}

IMPORTANT: Provide a selection for EVERY module. Use the protocol_selection function
call with analysis and selections (machine-checkable JSON only).
```

