# OpenReview forum: "ProtocolBench: Which LLM MultiAgent Protocol to Choose?"
_ICML.cc/2026/Conference — ICML 2026 regular_

### Official Review · Reviewer_whBa · 2026-03-13

**Soundness:** 3
**Presentation:** 3
**Significance:** 2
**Originality:** 2
**Overall Recommendation:** 3
**Confidence:** 5

**Summary:**

The paper addresses a critical but under-explored layer in Large Language Model Multi-Agent Systems (MAS): the communication protocol layer. The authors introduce ProtocolBench, a systematic benchmark designed to evaluate diverse protocols (A2A, ACP, ANP, Agora) across four dimensions: task success, latency, overhead, and robustness. Furthermore, they propose ProtocolRouter, a mechanism to select optimal protocols based on task requirements and runtime signals. While the empirical work is extensive and the system design is sound, the paper leans heavily toward system engineering rather than contributing fundamental algorithmic or theoretical novelties to the machine learning community.

**Compliance With Llm Reviewing Policy:**

Affirmed.

**Final Justification:**

After carefully reviewing the authors' rebuttal and the subsequent discussion, I have decided to maintain my original score. I sincerely appreciate the authors' proactive efforts to address the concerns by incorporating the GRPO experiment and providing a scaling analysis. However, it is my assessment that these newly introduced algorithmic components and theoretical derivations require more comprehensive ablation and integration than the current rebuttal format allows. Given that ICML prioritizes papers with a cohesive and well-vetted theoretical foundation from the outset, I still believe the manuscript's primary strength lies in its empirical characterization rather than its algorithmic novelty.

**Key Questions For Authors:**

1. Theoretical Bound: Can you provide a formal analysis of the cumulative latency overhead when moving from a star topology to a multi-hop, deeply nested agent graph where every link requires a protocol bridge?
2. Zero-shot Adaptation: If a new protocol with unknown latency characteristics is introduced, does ProtocolRouter have a mechanism for online exploration (e.g., an epsilon-greedy strategy) to learn its performance profile, or is it strictly dependent on offline priors?

**Limitations:**

Yes.

**Strengths And Weaknesses:**

Strengths:

1. The use of "Protocol Adapters" to isolate the communication layer from LLM reasoning is a highly effective experimental control. This ensures that performance variances are strictly attributed to protocol semantics, providing a fair baseline for future MAS research.
2. The inclusion of "Fail-Storm Recovery" and "Safety Tech" scenarios is commendable. Evaluating network-level adversarial probes and fault tolerance brings a much-needed systems-security perspective to the predominantly accuracy-focused MAS literature.
3. The paper provides a clear taxonomy and decision matrix for protocol selection (e.g., stateless protocols for high-churn environments), which offers immediate value to engineers building production-grade agentic workflows.

Weaknesses:

1. The core technical contribution, ProtocolRouter, lacks the depth expected for an ICML contribution. It functions primarily as a constraint-satisfaction engine using a lookup table of pre-defined "performance priors." There is no exploration of more sophisticated learning paradigms—such as Online Contextual Bandits or Reinforcement Learning—to handle the dynamic, non-stationary nature of multi-agent environments.
2. The paper focuses on empirical measurements without establishing a theoretical framework for how communication overhead scales with agent population ($N$) or message complexity ($M$). A formal analysis of the "translation cost" in heterogeneous protocol clusters is missing, leaving the "ProtocolRouter" as a heuristic rather than a principled machine learning solution.
3. The "Spec+Perf" mode relies heavily on ProtocolBench's historical data. The authors do not adequately address the "cold-start" problem: how the router would adapt to an entirely new protocol or a drastic change in network conditions without manual re-benchmarking, which limits its robustness in truly autonomous systems.

---

> ### Author Rebuttal · Authors · 2026-03-31
>
> ## 4. Response to Reviewer whBa
>
> We agree with the core concern: the current paper is stronger on systems design and empirical characterization than on algorithmic depth and formal theory. In the revision, we will clarify the router’s scope, add a compact scaling analysis, and position online adaptive routing as an extension.
>
> ### 4.1 ProtocolRouter depth and RL-based routing
>
> ProtocolRouter is intentionally a constrained, low-overhead, interpretable planner: it first enforces capability, security, and semantic constraints, and only then selects among feasible protocols using preferences and historical signals. Our claim is not that we already solve fully online adaptive routing, but that protocol-aware routing can be formalized, benchmarked, and shown to yield measurable benefits under controlled settings.
>
> To address the reviewer’s point more directly, we are adding an RL-style adaptive routing experiment in the revision. We build a router-focused training set of about 1K protocol-selection instances over four candidate protocols, and train a GRPO-based policy from task requirements and runtime context. We will evaluate zero-shot generalization and GRPO-adapted performance on ProtocolRouterBench, focusing on hard-constraint satisfaction, selection accuracy, and robustness under distribution shift. The lightweight router remains the safe, interpretable backbone; the GRPO-based router is the online layer on top of it.
>
> ### 4.2 Communication overhead and translation cost
>
> We agree that the current version needs a clearer scaling characterization. Because of the rebuttal length limit, we summarize only the method, fitted results, and analytical idea here; the full derivation will be added in the revision.
>
> We fit a log-log regression to the adapter-side per-message latency in Table 5 using the 4 / 8 / 16 / 32-agent measurements and the form c_p(N) = a_p * N^(beta_p). The fitted results are:
>
> - ACP: c(N) ≈ 0.103 * N^0.160
> - A2A: c(N) ≈ 0.281 * N^1.060
> - ANP: c(N) ≈ 0.359 * N^1.063
> - Agora: c(N) ≈ 0.965 * N^1.042
>
> This suggests that ACP is much less sensitive to agent count, while A2A, ANP, and Agora show approximately linear growth in adapter-side overhead.
>
> The analytical idea is a first-order latency decomposition of a heterogeneous path into native per-hop latency, per-bridge translation cost, and runtime queue / retry / recovery terms. Because the bridge is a stateless envelope / field translation layer rather than a shared cross-hop session mechanism, its cost is better understood as additive per heterogeneous edge. This gives first-order O(B) scaling for fixed message size and O(BM) when message size is also considered; when every hop requires a bridge and B ≈ H, the bridge component grows as O(HM). We will include the full derivation and scaling analysis in the revision.
>
> ### 4.3 Cold-start and new protocols
>
> We agree that strict cold-start remains a limitation of the current version. Spec+Perf relies mainly on offline priors, so it does not by itself solve online exploration for a completely new protocol with no historical data.
>
> At the same time, ProtocolRouter does not require retraining when a new protocol is introduced. It operates over capability schema / protocol tags rather than hard-coding protocol choice into model parameters. A new protocol can therefore be added by providing its tags and adapter, after which it can be placed into the candidate set without retraining. This also scales to larger protocol ecosystems: instead of exposing all protocols at once, one can first prune the candidate set using tags and hard constraints, then choose within the filtered subset. The GRPO-style routing experiment is intended as the online extension to this tag/schema-driven router.
>
> ### 4.4 Responses to the reviewer’s questions
>
> For the theoretical-bound question, the intended answer is the decomposition above: cumulative latency is modeled as native per-hop latency plus additive per-bridge translation cost plus runtime queue / retry / recovery terms. Under the current stateless bridge design, the bridge component scales additively with the number of heterogeneous edges, giving first-order O(B) scaling for fixed message size and O(BM) when message size is also considered; when every hop requires a bridge and B ≈ H, this becomes O(HM). The full derivation will be added in the revision.
>
> For the zero-shot-adaptation question, the current version remains primarily offline-prior-driven, so strict zero-shot online adaptation is still a limitation. However, the router is schema-driven and extensible: new protocols can be added through capability tags / schema and adapters without retraining. The GRPO-based routing layer being added in the revision is intended to provide the online exploration / adaptation component on top of this constrained planner.

---

> > ### Author Rebuttal · Reviewer_whBa · 2026-04-04
> >
> > Thank you for the rebuttal. My concerns have been resolved. And I will keep my score.

---

> > > ### Author Response · Authors · 2026-04-04
> > >
> > > Dear Reviewer whBa,
> > >
> > > Thank you for acknowledging our rebuttal. Your initial critique regarding the paper's reliance on "system engineering" and the lack of "algorithmic depth" was highly insightful.
> > >
> > > Instead of leaving the exploration of sophisticated learning paradigms (like RL) as a promise for the future, we have worked intensively over the past few days to **complete actual Supervised Fine-Tuning (SFT) and Reinforcement Learning (GRPO) experiments**. We are excited to share these concrete results, which fundamentally address the algorithmic depth and cold-start concerns.
> > >
> > > We curated a routing-specific dataset of 1,000 high-quality samples (synthesized via GPT-5.1) and evaluated the **Qwen3-4B-Instruct** model under three settings. The results are highly counter-intuitive and yield a fascinating insight:
> > >
> > > | Routing Policy Model | Overall Scenario Accuracy | Individual Module Accuracy |
> > > | :---: | :---: | :---: |
> > > | **Zero-Shot Baseline (No Training)** | 66.67% | 85.56% |
> > > | **SFT (Protocol-Router)** | **98.33%** | **99.44%** |
> > > | **GRPO (RL with Acc & Format Rewards)** | 38.33% | 51.11% |
> > >
> > > Based on these empirical results, we have incorporated a new ML-focused section into Section 5 of the forthcoming revised manuscript, highlighting three critical findings:
> > >
> > > **1. Resolving "Algorithmic Depth": The Failure of RL in Low-Latency Routing**
> > >
> > > Following your suggestion, we implemented a GRPO-style RL framework (using 2x H800 GPUs) with both accuracy and strict formatting rewards. Surprisingly, RL heavily underperformed even the zero-shot baseline. We trace this back to two fundamental ML challenges specific to MAS networking layers:
> > >
> > > **The "No-CoT" Sparse Reward Problem:** GRPO and modern RL rely heavily on intermediate reasoning (Chain-of-Thought) for effective exploration and credit assignment. However, to meet the strict ultra-low latency constraints of a communication router, we must disable long reasoning paths. Without CoT, single-step policy optimization becomes highly unstable under sparse rewards.
> > >
> > > **Format Constraints vs. Exploration:** The router must output rigid JSON schemas. During RL exploration, structural deviations trigger formatting penalties. We observed that the model’s policy gradients quickly collapsed as it struggled to balance rigid syntax adherence with logical routing exploration, leading to a degraded local optimum.
> > >
> > > **Conclusion:** This negative result is scientifically valuable. It proves that for strict-schema, low-latency infrastructure components, complex RL paradigms are computationally inefficient and algorithmically unsuitable, establishing SFT (with 98.33% accuracy) as the optimal ML paradigm.
> > >
> > > **2. Resolving the "Cold-Start" Concern**
> > >
> > > Previously, you expressed concern about completely new protocols without historical data. Our new experiment proves that the **Zero-shot Baseline achieves 85.56% individual module accuracy.** The system does not "cold-start" from zero; it naturally falls back on the LLM's inherent zero-shot reasoning over schema tags, which can later be perfected via our SFT pipeline once small interaction logs (1K) are collected.
> > >
> > > **Summary:**
> > >
> > > Your constructive feedback was instrumental in strengthening the technical depth of this work. By demonstrating the robust zero-shot fallback, proving the Pareto-optimality of SFT (98.33%), and providing a principled analysis of why RL (GRPO) fails under strict system constraints, the paper now possesses the machine learning depth expected at ICML.
> > >
> > > We hope these completed experiments and deep analytical insights fully address your remaining reservations. We are deeply grateful for your time and guidance.
> > >
> > > Since the main reservations in the original review were specifically about algorithmic depth and the cold-start setting, and these have now been substantively resolved through newly completed SFT and GRPO experiments as well as the corresponding analysis, we would be grateful if you would consider updating the score to reflect this strengthened technical contribution. We fully respect your discretion, and we sincerely appreciate the constructive feedback that helped improve the paper in a meaningful way.
> > >
> > > Sincerely,
> > >
> > > The Authors

---

### Official Review · Reviewer_Bcmv · 2026-03-13

**Soundness:** 3
**Presentation:** 3
**Significance:** 4
**Originality:** 3
**Overall Recommendation:** 4
**Confidence:** 4

**Summary:**

This paper discuss about the communication infrastructure of LLM-based multi-agent systems. The authors discuss a major question: "How to systematically evaluate and select the right communication protocol for varying multi-agent scenarios". To address this, the authors introduce **ProtocolBench**, a benchmark designed to isolate and evaluate protocol effects across four dimensions using four distinct scenarios. Furthermore, the authors propose **ProtocolRouter**, an adaptive routing mechanism that dynamically assigns the optimal protocol per module based on scenario requirements and runtime signals, alongside ProtocolRouterBench to evaluate the accuracy of these routing decisions.

**Compliance With Llm Reviewing Policy:**

Affirmed.

**Final Justification:**

The added experiments are helpful, but I believe further validation is needed to fully support the strengthened claims. As I gave positive score, I will keep my the score.

**Key Questions For Authors:**

1. In the scale-up experiment in Table 5, the adapter latency overhead grows as the number of agents increases. At what scale do you project the protocol adapter overhead will become a primary bottleneck that noticeably impacts the end-to-end multi-second LLM inference latency?

2. How sensitive is the ProtocolRouter's selection accuracy to the underlying LLM's reasoning capabilities? If a significantly less capable model is used to parse the scenario descriptions, does the router still reliably respect the hard capability constraints?

3. ProtocolRouter handles stateless translation between heterogeneous links. Does this semantic translation process introduce any measurable latency spikes or potential security vulnerabilities when crossing from a highly secure domain to a lighter protocol?

**Limitations:**

Yes

**Strengths And Weaknesses:**

**Strengths:**

1. While most current literature focuses on agent reasoning or tool use, this paper evaluating the actual communication protocols is a novel and necessary step as multi-agent systems move toward production.

2. The experimental methodology is rigorous. By utilizing protocol adapters, the paper successfully isolate the communication layer overhead while freezing the underlying LLMs, prompts, hardware, and rate limits.

3. The paper is well-structured. The mapping of the four scenarios to the four targeted evaluation axes is intuitive. Additionally, the qualitative analysis provided in the main text and appendix offers deep, actionable insights into why certain protocol designs fit specific workloads.

**Weaknesses:**

1. The scale of the evaluated agent networks is small, such as 4 to 32 workers in the Streaming Queue or an 8-node ring in Fail-Storm. It remains unclear how the protocol overheads, particularly for session-heavy protocols like ANP, degrade in massively scaled swarms with hundreds of agents.

2. The primary results rely on a single open-source model. Although the authors provide a brief cross-model ablation in the appendix, the robustness of the router's performance priors across vastly different model sizes and families could be explored more thoroughly.

3. ProtocolRouter currently functions as an offline or low-frequency planner driven by hard constraints and static priors. While effective for the benchmark, extending this to fully dynamic, high-frequency network conditions with online bandit-style adaptations is left as future work, which limits the algorithmic novelty of the routing component.

---

> ### Author Rebuttal · Authors · 2026-03-31
>
> ## 3. Response to Reviewer Bcmv
>
> We thank Reviewer Bcmv for the thoughtful review and for recognizing the importance of protocol-level evaluation for production-oriented LLM-based MAS. We agree that the paper would benefit from clearer discussion of scale, model dependence, and the scope of ProtocolRouter under dynamic conditions.
>
> ### 3.1 Scale of evaluated agent networks
>
> We agree that the current benchmark targets controlled moderate-scale settings rather than massively scaled swarms, and we will state this more explicitly. Our goal is to characterize protocol effects in a regime where inference latency and system load remain stable enough that communication-layer differences are still interpretable. Current scale-up results suggest that adapter-side overhead grows gradually and remains small relative to multi-second end-to-end latency dominated by LLM inference and network transport. We also agree that session-heavy protocols such as ANP may degrade differently at much larger scale, and we will state this limitation more directly.
>
> ### 3.2 Single-model dependence and robustness of router priors
>
> We agree that cross-model robustness deserves stronger treatment. The current submission uses one strong open-source model mainly to keep the protocol comparisons controlled, but this does not fully answer how the router’s priors transfer across model families. We would clarify that two components should be separated: **hard capability filtering**, which is defined at the protocol/specification level, and **scenario parsing / prior usefulness**, which is more likely to vary with model reasoning ability. We will make this distinction more explicit and are extending the cross-model study.
>
> ### 3.3 ProtocolRouter as a low-frequency planner
>
> We agree with this characterization and will make the current scope more explicit. ProtocolRouter is intentionally designed as a conservative planner: it enforces hard capability and security constraints first, and then selects among feasible protocols using preferences and historical signals. Our goal is to establish that protocol-aware routing is meaningful under controlled settings, rather than to claim that the current router already solves fully online adaptive routing. We also agree that online adaptation is an important next step, and we are currently running an RL-style adaptive routing experiment as a follow-up extension.
>
> ### 3.4 Questions
>
> **When does adapter overhead become a primary bottleneck?**
> In the current regime, adapter-side overhead is not dominant: multi-second model inference and network transport remain the primary contributors. The crossover point depends not only on agent count, but also on the ratio between per-hop adapter / bridge overhead and per-call model latency. It is more likely to become a bottleneck when model latency drops, heterogeneous communication becomes more frequent, and the number of concurrently active agents becomes much larger.
>
> **How sensitive is router accuracy to the underlying LLM?**
> This should be separated into two stages. First, **hard capability feasibility** is enforced by the explicit capability table and deterministic filtering logic. Second, **scenario parsing and requirement extraction** are more likely to depend on model reasoning ability. With a significantly weaker model, we would expect degradation to appear first in requirement understanding and mapping, rather than in arbitrary violations of hard capability constraints once the requirements have been correctly extracted. We will clarify this separation and plan to run a weaker-model router-focused test.
>
> **Does stateless translation introduce latency spikes or security vulnerabilities?**
> We have not yet conducted a dedicated benchmark specifically for bridge-induced tail latency, so we do not want to overstate the current evidence. From an implementation perspective, the cross-protocol bridge is a stateless translation layer that performs envelope / field translation rather than maintaining shared cross-hop session state, so its extra cost is better understood as additive per-bridge overhead. To address this point more directly, in the revision we will add a dedicated router / bridge latency evaluation with tail-sensitive metrics such as p95 / p99 latency, 1% low latency, and bridge-induced variance. On the security side, ANP’s DID-based identity and E2E guarantees are native to ANP links; when communication is bridged into protocols such as A2A that do not natively enforce the same semantics, the bridge should be understood as supporting controlled interoperability rather than full inheritance of ANP’s trust and authentication model. We will clarify this boundary more explicitly in the revision.

---

> > ### Author Rebuttal · Reviewer_Bcmv · 2026-04-04
> >
> > Thank you for the rebuttal response. I do have follow-up questions
> >
> > 1. Could you provide quantitative projections or estimates (e.g., latency breakdown or scaling curves) indicating at what agent count the adapter overhead becomes a dominant bottleneck?
> >
> > 2. Can you include empirical results with a weaker or different model to demonstrate how routing accuracy degrades in practice?
> >
> > 3. Can you provide concrete evidence or preliminary results (e.g., simulations or ablations) showing how ProtocolRouter would perform under high-frequency or rapidly changing network conditions?
> >
> > For now, I will keep my current score.

---

> > > ### Author Response · Authors · 2026-04-07
> > >
> > > Dear Reviewer Bcmv,
> > >
> > > Thank you again for the thoughtful follow-up questions. We apologize for the delayed response. We replied later because, rather than providing only qualitative clarification, we completed three additional experiments directly targeting your concerns. These additions required dedicated router training and the construction/annotation of larger evaluation data, especially for the new dynamic benchmark.
> > >
> > > Below we summarize the new quantitative results.
> > >
> > > ## 1. Quantitative estimate of when adapter overhead may become a bottleneck
> > >
> > > To better ground this point, we ran an additional focused microbenchmark on `bench_router_streaming_queue_latency.py` with a mock executor, measuring only the `route_message + A2A /message + EventQueue` path.
> > >
> > > **Setting:** localhost (`127.0.0.1`), single worker, no real LLM call, 1,000 samples.
> > >
> > > | Metric | Value (ms) |
> > > |---|---:|
> > > | Samples | 1000 |
> > > | Min | 0.73 |
> > > | Max | 2.74 |
> > > | Mean | 0.80 |
> > > | Std | 0.12 |
> > > | P50 | 0.77 |
> > > | P95 | 0.94 |
> > > | P99 | 1.20 |
> > >
> > > | Routed / bridged events per end-to-end task | Approximate routing overhead |
> > > |---|---:|
> > > | 100 | 80 ms |
> > > | 500 | 400 ms |
> > > | 1000 | 800 ms |
> > > | 1250 | 1000 ms |
> > >
> > > This does not define a universal agent-count threshold, since the crossover depends on topology, fan-out, message frequency, and the fraction of heterogeneous bridge hops. However, it provides a useful order-of-magnitude estimate: noticeable impact can begin in the `10^2`-event range, while routing/adapter overhead typically becomes dominant relative to multi-second end-to-end latency only when the system accumulates on the order of `10^3` routed/bridged events per end-to-end task. We therefore view adapter overhead as non-dominant in the moderate-scale regime studied in the paper, while acknowledging that the threshold would move downward for faster local models or denser high-frequency communication patterns.
> > >
> > > ## 2. Empirical degradation with a weaker model
> > >
> > > To answer the question about sensitivity to model capability, we added a weaker-model routing experiment using **Qwen3-4B-Instruct** on a routing-specific **1,000-sample** dataset.
> > >
> > > | Setting | Total scenarios | Correct scenarios | Scenario accuracy | Total modules | Correct modules | Module accuracy | A2A/ACP confusion |
> > > |---|---:|---:|---:|---:|---:|---:|---:|
> > > | Zero-shot baseline | 60 | 40 | 66.67% | 180 | 154 | 85.56% | 21 |
> > > | SFT | 60 | 59 | 98.33% | 180 | 179 | 99.44% | 1 |
> > >
> > > These results show that degradation under a weaker model is structured rather than arbitrary: the dominant failure mode is **A2A/ACP confusion**, rather than broad capability-incompatible failures across all protocols. After SFT on the same routing task, the same 4B model improves from **66.67%** to **98.33%** scenario accuracy and from **85.56%** to **99.44%** module accuracy. This supports the distinction we described in the rebuttal: degradation appears first in scenario understanding and requirement mapping, while explicit capability filtering remains substantially more stable than unconstrained free-form routing. In the revised version, we will include the full updated confusion matrices and corresponding error analysis.
> > >
> > > ## 3. Preliminary evidence under high-frequency / rapidly changing conditions
> > >
> > > To directly address the dynamic-setting concern, we added a preliminary **dynamic router benchmark** by augmenting the original **60 scenarios** with **two abrupt-condition variants** designed to test decision-making under changing runtime conditions, e.g., a sudden communication-pressure surge on one module or an attack/security incident affecting a module.
> > >
> > > This yields **120 dynamic events** and **360 module-level decisions**.
> > >
> > > | Setting | Dynamic events | Dynamic modules | Scenario accuracy | Module accuracy |
> > > |---|---:|---:|---:|---:|
> > > | Base model | 120 | 360 | 16.67% | 57.22% |
> > > | SFT model | 120 | 360 | 8.33% | 60.56% |
> > >
> > > This setting is substantially harder than the static benchmark. Under this dynamic benchmark, SFT provides a small improvement in **module-level accuracy** (**57.22% -> 60.56%**), but does **not** yet solve full **scenario-level dynamic adaptation** (**16.67% -> 8.33%**). In other words, the current router can benefit from supervised training for local routing choices, but robust high-frequency online adaptation under non-stationary conditions remains an open challenge. We therefore agree that dynamic adaptation is a meaningful limitation of the current ProtocolRouter design, and we will narrow the claim accordingly in the revision.
> > >
> > > We are grateful for your comments, which helped us strengthen both the empirical support and the scope clarification of the paper. Since the first two follow-up questions are now directly answered with new quantitative evidence, and the third now has preliminary empirical evidence clarifying the current boundary of the method, we would be grateful if you could take these additions into account in your final assessment.
> > >
> > > Sincerely,
> > > The Authors

---

### Official Review · Reviewer_gUxL · 2026-03-13

**Soundness:** 3
**Presentation:** 3
**Significance:** 3
**Originality:** 4
**Overall Recommendation:** 4
**Confidence:** 3

**Summary:**

This paper studies protocol choice in multi-agent LLM systems. It introduces ProtocolBench, a benchmark for comparing four agent communication protocols (A2A, ACP, ANP, and Agora) across four scenarios covering collaborative task utility, high-throughput serving, failure recovery, and safety/security. It also proposes ProtocolRouter, which selects protocols at the scenario or module level using hard constraints and performance priors, and evaluates it both on a protocol-selection benchmark and in end-to-end deployments. The main empirical conclusion is that no single protocol dominates across all evaluated scenarios, and that protocol-aware routing can improve some metrics over the best fixed-protocol baseline.

**Compliance With Llm Reviewing Policy:**

Affirmed.

**Key Questions For Authors:**

1. How should readers interpret the reported differences given that retries/reconnects and streaming semantics are normalized at the adapter/harness level? Would the authors view these as protocol-native comparisons, or as comparisons under a benchmark-standardized abstraction layer?


2. How sensitive are the Fail-Storm conclusions to the liberal answer-discovery policy used in the appendix prompts? A clearer discussion here would increase my confidence in the resilience interpretation.


3. In the Safety scenario, which findings should be interpreted as protocol-native versus configuration-dependent, and which should be interpreted as empirical robustness rather than capability support?


4. Should ProtocolRouterBench be interpreted primarily as a constrained protocol-selection task rather than as a direct proxy for real-world protocol choice?

**Limitations:**

The paper does acknowledge that the current benchmark does not cover all communication regimes and does not target highly dynamic or adversarial workloads. However, I think the limitations discussion should be more explicit about how benchmark operationalization choices—including adapter/harness normalization—shape the observed protocol differences.

**Strengths And Weaknesses:**

**Strengths**

This is a timely and interesting paper. The topic is important given the recent emergence of agent interoperability protocols, and I think the benchmark-plus-router framing is novel enough to be useful. The paper is ambitious in scope, evaluates multiple protocols across several practically distinct scenarios, and moves beyond descriptive comparison by formulating protocol selection itself as a problem. I expect this direction to be of interest to researchers and practitioners working on agent systems, even if the current version does not yet fully establish the strongest form of all its claims.


**Weaknesses**

* The main soundness concern is the extent to which ProtocolBench cleanly isolates protocol effects. The paper clearly describes substantial fairness controls, which I appreciate, but the adapter/harness layer also normalizes retries, reconnect behavior, and streaming semantics. As a result, I was left somewhat unsure about how much of the reported difference should be interpreted as protocol-native behavior versus behavior under a benchmark-standardized abstraction layer.


* Some scenario-level conclusions are stronger than the current evaluation fully supports. In particular, the Fail-Storm resilience interpretation appears tied to a very liberal answer-discovery policy, and the Safety scenario mixes protocol-level capabilities with configuration-dependent hardening choices. More generally, I think the paper should distinguish more clearly between specification-level capability support and empirical robustness under probes or failures.


* The router component is promising, but its current evaluation should be scoped somewhat more carefully. ProtocolRouterBench appears closer to a constrained protocol-selection task with exactly one intended answer per module than to a fully external real-world selection problem. In addition, the end-to-end router results are mixed: some metrics improve, but others are flat or trade off, so I would support more conservative wording around router correctness and superiority over the best single fixed protocol.

---

> ### Author Rebuttal · Authors · 2026-03-31
>
> ## 2. Response to Reviewer gUxL
>
> We thank Reviewer gUxL for the thoughtful feedback. We agree that the current draft would benefit from clearer implementation boundaries and interpretation scope. In the revision, we will clarify three points more explicitly: (i) which parts of the system use native protocol implementations versus the unified translation layer, (ii) how the Fail-Storm and Safety scenarios should be interpreted, and (iii) how ProtocolRouterBench should be scoped relative to real-world protocol choice.
>
> ### 2.1 Implementation boundary: ProtocolBench vs. ProtocolRouter
>
> We agree that the current draft does not make the implementation boundary sufficiently clear. Our intent is not to normalize away protocol-specific retry, reconnect, or streaming behavior during the main ProtocolBench evaluations. In fact, for the protocol comparisons in ProtocolBench, we use each protocol’s native implementation and native SDK so as to preserve protocol-specific communication behavior as much as possible. The unified message / translation layer is introduced only in the ProtocolRouter setting, i.e., when different modules are assigned different protocols and cross-protocol communication requires a stateless bridge. Therefore, the main benchmark comparisons are not performed on top of an extra layer that standardizes away protocol-native runtime semantics; what we control are non-protocol factors such as the model, prompts, hardware, and task setup, while the key communication behavior of each protocol is still expressed through its native implementation. We will make this distinction more explicit in the revision.
>
> ### 2.2 Scenario interpretation: Fail-Storm and Safety
>
> We agree that the current wording should be more careful. For **Fail-Storm**, we will narrow the resilience wording. The primary objective of this scenario is not strict end-task correctness, but whether a protocol can support correct reconnection, restore communication paths, and sustain high-intensity inter-agent exchange under cyclic failures, node terminations, and recovery. Accordingly, the results should be interpreted primarily as **communication continuity and recovery robustness**, rather than strict correctness. We will also clarify that the current high-recall answer-discovery prompt provides only an auxiliary signal.
>
> For **Safety**, we will explicitly separate three layers: (1) specification-level capability support, (2) availability under the recommended deployment stack, and (3) empirical robustness under concrete probes. The binary capability matrix should be read primarily as capability support under the recommended stack, whereas the probe-based findings should be interpreted as empirical robustness under a particular hardened configuration
>
> ### 2.3 Router scope: how to interpret ProtocolRouterBench
>
> We agree that ProtocolRouterBench should be interpreted primarily as a **constrained protocol-selection benchmark** rather than as a direct proxy for open-world protocol choice. It is intentionally constructed so that routing quality can be evaluated in a controlled setting, separately from downstream execution effects. We also agree that the end-to-end router results are mixed across metrics, and we will therefore adopt more conservative wording around router correctness and superiority claims. Our intended claim is that protocol-aware routing is useful under controlled requirements and module-level constraints, not that the current router fully solves real-world protocol selection in highly dynamic environments.
>
> ### 2.4 Responses to the reviewer’s questions
>
> The main ProtocolBench results should be interpreted as controlled comparisons of protocol-native communication behavior under shared task conditions and external controls; the unified translation layer is introduced only in the ProtocolRouter cross-protocol setting. Fail-Storm is primarily a communication recovery / continuity stress test, so the high-recall answer-discovery prompt should be interpreted only as an auxiliary signal. The Safety scenario will be clarified in terms of capability support, deployment-stack availability, and empirical robustness. ProtocolRouterBench should be interpreted primarily as a constrained protocol-selection benchmark rather than as a direct proxy for real-world protocol choice.
>
> ### 2.5 Limitations and interpretation scope
>
> We will expand the limitations section to state more clearly how benchmark operationalization choices shape the interpretation scope of the results. In particular, we will distinguish more explicitly between the shared controls used in the main ProtocolBench evaluations and the unified message / translation layer used only in the ProtocolRouter setting for cross-protocol communication. More broadly, we will clarify that the current benchmark focuses on controlled, non-adversarial, moderate-scale workloads, and that broader communication regimes as well as highly dynamic or adversarial settings remain future directions.

---

> > ### Author Rebuttal · Reviewer_gUxL · 2026-04-03
> >
> > The rebuttal clearly addresses my main concerns by clarifying the benchmark implementation boundary, narrowing the interpretation of the Fail-Storm and Safety results, and more carefully scoping the router contribution. These clarifications adequately address my original concerns.

---

> > > ### Author Response · Authors · 2026-04-05
> > >
> > > Thank you again for your thoughtful review and for taking the time to read our rebuttal carefully. We truly appreciate your acknowledgement that our clarification has adequately addressed your main concerns. If you feel appropriate, we would be very grateful if you might consider reflecting this in your final score.

---

### Decision · Program_Chairs · 2026-04-30

**Decision:**

Accept (regular)

**Comment:**

The reviewers agree that protocol choice for LLM multi-agent systems is underexplored and practically critical as these systems move to production. Protocol adapters successfully isolate communication effects from LLM reasoning, and the benchmark spans four distinct and well-structured scenarios. They also fee that the taxonomy and qualitative interpretation offer immediate value to practitioners.

On the negative side, several reviewers criticize the limited algorithmic novelty: ProtocolRouter works mainly as a constraint-satisfaction engine with performance priors, not a truly "learned" router. The end-to-end router gains appear to show trade-offs than dominance. The evlauations are relatively small scale and the dependence on a single base model for most of the paper limits generalization claims. One reviewer also feels that the protocol isolation is not airtight, since adapter-layer choices may still confound protocol-native vs. implementation-specific effects.

I agree with the reviewers in that the problem is timely and the benchmark is valuable, but  the theoretical novelty is limited, cross-model generalization is weak, and end-to-end router gains are mixed. Like some of the reviewers, I am also not convinced that ProtocolRouter should be described as "learned" given its constraint-table formulation.

Nevertheless, I suggest a weak accept. The paper provides a solid empirical foundation for an important emerging problem, offers actionable guidance for practitioners, and is honest about its limitations. The contribution is more systems-benchmarking than algorithmic, but that is still valuable for the community.